# Drug Discovery Targeting Focal Adhesion Kinase (FAK) as a Promising Cancer Therapy

**DOI:** 10.3390/molecules26144250

**Published:** 2021-07-13

**Authors:** Xiao-Jing Pang, Xiu-Juan Liu, Yuan Liu, Wen-Bo Liu, Yin-Ru Li, Guang-Xi Yu, Xin-Yi Tian, Yan-Bing Zhang, Jian Song, Cheng-Yun Jin, Sai-Yang Zhang

**Affiliations:** 1Key Laboratory of Advanced Drug Preparation Technologies (Ministry of Education), Institute of Drug Discovery & Development, School of Pharmaceutical Sciences, Zhengzhou University, Zhengzhou 450001, China; summer_pxj@163.com (X.-J.P.); LXJ120WYY@163.com (X.-J.L.); liuyuan9720@163.com (Y.L.); 17698567571@163.com (W.-B.L.); zhangyb@zzu.edu.cn (Y.-B.Z.); 2School of Basic Medical Sciences, Zhengzhou University, Zhengzhou 450001, China; 17603868158@163.com (Y.-R.L.); ygx19990110@126.com (G.-X.Y.); txyi626@163.com (X.-Y.T.); 3Henan Institute of Advanced Technology, Zhengzhou University, Zhengzhou 450001, China; 4State Key Laboratory of Pharmaceutical Biotechnology, Nanjing University, Nanjing 210023, China

**Keywords:** focal adhesion kinase, anticancer activity, diaminopyrimidines, triazine, PROTAC

## Abstract

FAK is a nonreceptor intracellular tyrosine kinase which plays an important biological function. Many studies have found that FAK is overexpressed in many human cancer cell lines, which promotes tumor cell growth by controlling cell adhesion, migration, proliferation, and survival. Therefore, targeting FAK is considered to be a promising cancer therapy with small molecules. Many FAK inhibitors have been reported as anticancer agents with various mechanisms. Currently, six FAK inhibitors, including **GSK-2256098** (Phase I), **VS-6063** (Phase II), **CEP-37440** (Phase I), **VS-6062** (Phase I), **VS-4718** (Phase I), and **BI-853520** (Phase I) are undergoing clinical trials in different phases. Up to now, there have been many novel FAK inhibitors with anticancer activity reported by different research groups. In addition, FAK degraders have been successfully developed through “proteolysis targeting chimera” (PROTAC) technology, opening up a new way for FAK-targeted therapy. In this paper, the structure and biological function of FAK are reviewed, and we summarize the design, chemical types, and activity of FAK inhibitors according to the development of FAK drugs, which provided the reference for the discovery of new anticancer agents.

## 1. Introduction

Receptor tyrosine kinases (RTK) and non-receptor tyrosine kinases play vital roles in extracellular signals and process of cancer cell migration, proliferation, and survival, which usually are overexpressed in many human cancer cells [1,2]. Therefore, targeting RTKs and non-receptor tyrosine kinases has been considered to be a promising therapy for the treatment of human cancers [3].

FAK is one of cytoplasmic non-receptor protein tyrosine kinase, which is a member of the focal adhesion complex family [4,5]. FAK is a 125 kDa protein that is recruited in the early stages of cell adhesion and phosphorylated to mediate the formation of local adhesion [6,7,8,9]. FAK gene exists on chromosome 8q24 and encodes 1052 amino acid residues and its protein structure consists of three parts: *N*-terminal region, kinase region, and *C*-terminal region [9].

FAK was first discovered in 1992, which is related to integrin signal transduction [4]. FAK is considered to be a key regulator of growth factor receptor and integrin-mediated signaling, which controls the basic processes of human normal or cancer cells through its kinase activity and scaffold function [10,11]. FAK overexpression is detected in various tumor types, including prostate cancer, breast cancer, lung cancer, ovarian cancer, and neck cancer, and FAK is also associated with the poor prognosis of cancer patients [12,13]. Besides, it is related to unregulated proteins in cancer, providing a perfect environment for FAK-specific treatment [14]. FAK regulates tumorigenic and metastatic potential through a highly coordinated signal network, thereby promoting the occurrence of malignant tumors. These signal networks coordinate a series of cellular processes, such as cell proliferation, survival, invasion, migration, angiogenesis, and the regulation of cancer stem cell activity [5,15,16]. Different FAK antisense oligonucleotides [17], dominant-negative FAK *C*-terminal domain, FAK-CD or FRNK or FAK-siRNA [18,19] can lead to decreased cell survival, growth inhibition, or apoptosis. In recent years, FAK has been considered as a new potential tumor-treatment target [20].

So far, some FAK inhibitors have been identified and proved to be effective inhibitors that inhibit the tumor growth and metastasis in several preclinical and clinical models [21,22,23,24]. **TAE-226** is a typical FAK inhibitor (IC_50_ = 5.5 nM), which has strong antiproliferative and anticancer potency against several types of cancers in vitro and in vivo [21]. **GSK2256098** is a highly potent, ATP-competitive, and reversible FAK inhibitor [25], which is currently underway in clinical trials as a monotherapy or in combination with other drugs for patients with brain tumors or other cancers [26]. **VS-6063** is a highly effective second-generation FAK inhibitor. Phase II clinical trials of **VS-6063** in patients with KRAS mutant non-small cell lung cancer have been completed [27]. **VS-6062** has now completed the first phase of research. **VS-6062** not only exhibits significant antiproliferation potency of cancer cells but also plays a new role in changing tumor microenvironment [28]. **VS-4718** significantly affects cell viability by inhibiting FAK activity and locking Y397 phosphorylation [23,24]. Another novel FAK inhibitor, **CEP-37440**, is currently in clinical trials for the treatment of cancers [22] (Table 1). The proteolytic targeting chimera (PROTAC) that can induce FAK degradation by using FAK kinase inhibitors as a “warhead” also has been developed [29]. A large number of studies have provided convincing evidence that FAK inhibitors could be used as chemical-targeted drugs to induce growth inhibition and apoptosis of cancer cells [30,31,32]. This article briefly introduces the structure and biological functions of FAK. Aiming at the development of FAK drugs, it focuses on the design, chemical types, and activities of FAK drugs to provide references for the development of new antitumor drugs.

## 2. FAK Structure and Function

The FAK protein molecule can be roughly divided into *N*-terminal FERM region, middle kinase region, *C*-terminal focal adhesion target (FAT), and proline-rich regions (PRR1, PRR2, and PRR3). FAK has 6 tyrosine sites that can be phosphorylated: Y397, Y407, Y576, Y577, Y861, and Y925, all of them are the key parts in FAK’s signal transduction. Y397 and Y407 sites are located at the *N*-terminus, Y576 and Y577 are located in the activation loop of the kinase region, and Y861 and Y925 are located at the *C*-terminus. Y397 is an autophosphorylation site of FAK, which provides a high-affinity-binding site for Src family proteins and plays a vital role in downstream signaling pathways [16].

There are binding sites for a variety of proteins at the *N*-terminus, such as signal transduction proteins, cytoskeleton proteins, and integrin β subunits [33]. The *N*-terminal FERM domain is composed of three closely related subdomains F1, F2, and F3, forming a three-leaf clover-like structure [34] (Figure 1). Among of them, F1 and F2 subdomains can interact with p53 and hinder the apoptosis of tumor cells; the F2 subdomain can regulate kinase-independent activities and mediate cell survival; the F3 subdomain can interact with Mdm-2 to enhance ubiquitination of p53 [33]. The kinase domain of FAK also called the catalytic domain, has a highly conserved amino acid sequence that can phosphorylate corresponding amino acid residues in proteins such as PI3K, growth factor receptor-binding protein 2 (GRB2), Src, and Cas. The focal adhesion targeting region at the *C*-terminus could interact with focal adhesion proteins, including paxillin [35], Talin [36], and Vinculin [37]. FAT can also bind to the F2 leaf in the FERM domain, leading to the dimerization of FAK, which is of great significance to the activation and nuclear localization of FAK [38]. PRR1 is located between the FERM region and the kinase region and has a binding site to the SH3 domain, which can bind to proteins such as Src. PRR2 and PRR3 near the *C*-terminus can mediate the interaction between FAK and protein molecules containing SH3 domains.

FAK shows kinase-dependent enzyme function and a kinase-independent scaffold function, which play important roles in regulating the survival, proliferation, migration, and angiogenesis of cancer cells. FAK is overexpressed in many human cancers, including glioma, breast, colon, prostate, and other cancers [10,16,39]. Inhibition of FAK has been shown to inhibit tumor progression and metastasis.

## 3. The Discovery of FAK Inhibitors and Their Therapeutic Significance

Because of its important roles in human cancer cells, targeting FAK inhibitors is emerging as a promising target for cancer therapy with small molecules. Currently, six FAK inhibitors are undergoing clinical and many FAK inhibitors have been reported with strong inhibitory activity of FAK and significant antiproliferation potency against cancer cells (Table 1). In this section, we summarized the progress for the development of FAK inhibitors.

### 3.1. FAK Inhibitors in Clinical Trials

**GSK2256098** is a highly effective and selective FAK inhibitor (IC_50_ = 0.4 nM). **GSK2256098** significantly inhibited FAK-Y397 phosphorylation in A549, OVCAR8, and U87MG cells (IC_50_ = 12, 15, and 8.5 nM, respectively). **GSK2256098** inhibited FAK Y397 phosphorylation and then suppressed FAK-related Akt/ERK activation, which led to the decrease of cell viability, anchorage-independent growth, and motility in a dose-dependent manner in pancreatic ductal adenocarcinoma cells [40]. The results of the phase I trial of **GSK2256098** in sixty-two patients with advanced solid tumors suggested **GSK2256098** had an acceptable safety profile and showed clinical activity in patients with mesothelioma. A phase Ib trial study indicated that **GSK2256098** combined MEK inhibitor trametinib in the treatment of patients with advanced solid tumors could increase the exposure of trametinib and has an acceptable safety profile [26].

**VS-6063** is another FAK and Pyk2 dual inhibitor with more effective activity against FAK and Pyk2 than that of **VS-6062**. **VS-6063** inhibited FAK-Y397 phosphorylation in a time and dose-dependent manner in ovarian cancer cells. The combination of **VS-6063** with paclitaxel could significantly exhibit the antiproliferation effects and increased apoptosis in ovarian cancer cells [24]. Notably, **VS-6063** could overcome YB-1-mediated paclitaxel resistance by an AKT-dependent pathway. **VS-6063** showed an acceptable safety profile in forty-six patients with advanced solid tumors according to the results of phase I trial [27]. The phase II trial, multicenter cohorts, open-label, single-arm study of **VS-6063** treatment in heavily patients with KRAS mutant NSCLC indicated **VS-6063** monotherapy demonstrated modest clinical activity [41].

**VS-6062** is an ATP-competitive FAK and Pyk2 dual inhibitor, which displayed significant inhibitory potency of FAK and Pyk2 (IC_50_ = 1.5 and 14 nM, respectively) [42]. **VS-6062** potently blocked FAK-Y397 phosphorylation in a dose-dependent manner in Ewing sarcoma cell lines which resulted in the downregulation of its downstream targets AKT/mTOR and CAS activity. **VS-6062** could suppress the growth and colony formation, and induced apoptosis in Ewing sarcoma cell lines [43]. The combined treatment of **VS-6062** with sunitinib could potently inhibit angiogenesis and tumor proliferation on liver solid tumor models [44]. In addition, **VS-6062** has been reported to be an effective antimetastatic and antiangiogenic agent, which could potently suppress EOC and endothelial cell migration and endothelial cell tube formation in vitro [45]. The results of phase I trial of **VS-6062** in ninety-nine patients with advanced solid tumors indicated that **VS-6062** showed an acceptable safety profile and displayed time- and dose-dependent nonlinear PK [46].

**VS-4718** is a highly effective and selective FAK inhibitor (IC_50_ = 1.5 nM) [47]. **VS-4718** displayed potent inhibition activity with IC_50_ values ranging from 0.25 μM to 3.53 μM in vitro in a pediatric preclinical testing program (PPTP) and exhibited excellent tolerance in PPTP preclinical xenografts in vivo [48]. In addition, it has been reported that **VS-4718** could antagonize the multidrug resistance in ABCB1- and ABCG2-overexpressing cancer cells through competitively interacting with ABCB1 and ABCG2 and affecting the activity of ABCB1 and ABCG2, which indicated that **VS-4718** combines with other anticancer drugs may antagonize the multidrug resistance [49,50,51,52]. The phase 1 clinical trial of **VS-4718** combined with paclitaxel and gemcitabine for the treatment of advanced pancreatic cancer has been completed [53].

**CEP-37440** is a potent FAK and ALK dual inhibitor with strong and selective inhibition activity of FAK and ALK (IC_50_ = 2.0 and 3.1 nM, respectively). **CEP-37440** effectively suppressed the proliferation of Sup-M2 and Karpas-299 cells (NPM-ALK+ ALCL cell lines) with IC_50_ values 84 and 131 nM, respectively. **CEP-37440** displayed favorable in vitro ADME properties and acceptable oral bioavailability in CD-1 mice, Sprague−Dawley rats, and cynomolgus monkeys. In addition, **CEP-37440** is effective in preclinical models of FC-IBC02, SUM190, and KPL4 with GC_50_ values of 91, 900, and 890 nM, respectively [22,54]. **CEP37440** is currently undergoing a human clinical phase I trial.

**CEP-37440** was discovered by the optimization of lead compound **1** which is an ALK inhibitor containing a diaminopyrimidine core motif [55,56,57] (Figure 2A). When morpholine was switched to *N*-methyl piperazine (compound **2**), the stability of compound **2** in rat liver microsomes was significantly improved without eroding the potency or selectivity. Based on these promising results of compound **2**, the strategy is to further increase the inhibitory potency of ALK, maintaining the activity of auxiliary FAK, and improve the metabolic stability of human liver microsomes. Changing the substituent of *N*-methyl on the piperazine to a hydroxyethyl substituent (compound **3**) showed more potent inhibitory activity against enzymes and cells. Based on compound **3**, changing the bicycloheptene system to 2-(*N*-methylmethanesulfonamide) benzene ring gave compound **4**, which exhibited similar ALK enzyme and cells inhibitory potency. When the 2-*N*, *N*-dimethylsulfonamide substituent was replaced with *N*-methylamide substituent (**CEP-37440**), it displayed acceptable oral bioavailability (47%) and demonstrated acceptable in vitro liver microsome stability across species (rat and human *t*_1/2_ > 40 min) and also exhibited great inhibitory potency of ALK and FAK. The molecular docking results also explained that compound **CEP-37440** exhibited great dual ALK/FAK potency (Figure 2B) [22,58].

### 3.2. Pyrimidine FAK Inhibitors

In 2006, Novartis researchers first reported an ATP-competitive FAK inhibitor named as **TAE-226** (Figure 3A). **TAE-226** is a highly effective and orally active FAK and Pyk2 dual-target inhibitor [59]. **TAE-226** exhibited a wide spectrum of antiproliferation activity against various human cancer cells with a mean GI_50_ value of 0.76 μM [60]. **TAE-226** potently inhibited FAK-Y397 autophosphorylation and blocked FAK-mediated downstream signaling pathways [61]. **TAE-226** showed a strong antitumor effect through suppressing migration and metastasis, and induced increased apoptosis in a variety of cancer cells, such as glioma, breast, ovarian, and neuroblastoma cells [62]. The crystal structure of **TAE-226** complexed with the kinase domain of FAK has been determined (Figure 3B), which clearly demonstrates the interaction of **TAE-226** with FAK [63]. However, in consideration of its serious side effects on sugar metabolism, **TAE-226** never entered clinical trial but is usually compared with other newly developed FAK inhibitors as a positive drug [64].

**PF-573228** (Figure 4) is the first selective and potent ATP-competitive FAK inhibitor developed by Pfizer [65], which potently suppressed the FAK activity with an IC_50_ of 4 nM and the phosphorylation of FAK on Tyr397 with IC_50_ values of 30–100 nM in F-G, SKOV-3 and PC3 cells. **PF-573228** inhibited the chemotactic and haptotactic migration, resulting in concomitant with the inhibition of focal adhesion turnover in REF52 cells. **PF-573228** could not inhibit the cell growth or induce cell apoptosis in REF52 and PC3 cells [66]. Based on the structural modifications of **PF-573228**, another two analogs **VS-6062** and **VS-6063** developed by Pfizer have been used in clinical trials.

2-Anilino-4-(benzimidazol-2-yl) pyrimidine **9** is a multi-kinase inhibitor with inhibitory potency of FAK, PLK1, Aurora B, and VEGF-R2 with IC_50_ values of 3.4, 1.2, 6, and 7.2 μM, respectively (Figure 5). Compound **9** also exhibited potent inhibition growth of SW-620, MDA-MB-435, OVCAR-3, CAKI-1, and MCF-7 cells at micromolar concentrations (IC_50_ values = 0.62, 0.19, 0.93, 0.86 and 0.47 μM, respectively) [67].

Farand’s group reported novel macrocycle diphenylpyrimidine derivatives as FAK inhibitors based on the lead compound **VS-6062** in 2016 (Figure 6). The representative compound **10** showed strong antagonism against FAK and Pyk2 with IC_50_ values of 4.34 nM and 0.84 nM, respectively [68]. Among these compounds, compound **10** showed the best metabolic stability in the analysis of human microsomes.

Sulfonamide functional groups exist in many active compounds and exhibit antibacterial, anticancer, and antiviral activities [69,70,71]. Previous studies have shown that the introduction of functional sulfonamides into the C-2 aniline portion of the pyrimidine core could increase the inhibitory potency of Bruton’s tyrosine kinase (BTK), which is a successful and effective drug target for the treatment of B lymphocytic leukemia [72]. Therefore, Ma’s group synthesized novel diphenylpyrimidine derivatives (Figure 7) through the introduction of sulfonamide functional groups based on the famous FAK inhibitor **TAE-226** to improve the anti-FAK activity and biological activity in 2017 [73].

These new compounds showed moderate anti-FAK enzyme activity compared to **TAE-226** (IC_50_ = 5.5 nM). The most active compound **18** effectively inhibited FAK activity with an IC_50_ value of 86.7 nM. It is worth noting that compound **18** exhibited significantly inhibitory potency against pancreatic cancer cells ASPC-1 and PANC-1 with IC_50_ values of 3.92 and 0.53 μM, respectively, which were lower than that of **TAE-226** (IC_50_ = 6.73 and >20 μM, respectively). Compound **18** also promoted the apoptosis of pancreatic cancer cells in a dose-dependent manner. Molecular docking results showed compound **18** bonded to the FAK enzyme tightly and the interaction with the FAK enzyme was similarly to **TAE-226,** which might explain compound **18** shows potent anti-FAK activity (Figure 8B). These results provide a theoretical basis for further optimizing the structure of novel sulfonamide compounds as FAK inhibitors.

In general, most of the potent FAK inhibitors have pyrimidine templates and *N*-methyl formamide substituted anilines, which are necessary to interact with amino acids Cys502, Asp564, and Leu567 of FAK protein to interfere with FAK activity [74,75,76]. Phosphorylated functional groups are very effective in improving the water solubility and biological activity of anticancer agents [77,78,79]. Ma’s group discovered that the introduction of phosphoryl groups on the C-2 aniline side chain of the pyrimidine core could improve its inhibitory activity of BTK and antiproliferation activity against B-cell leukemia cells [80,81,82,83]. In 2017, they designed and synthesized a series of novel diphenylpyrimidine analogues containing phosphamides as FAK inhibitors [84] (Figure 9).

Most of the phosphamide-containing diphenylpyrimidine analogs showed good activity against FAK, which were equivalent or showed stronger inhibition than that of **TAE-226** (IC_50_ = 5.5 nM). Compound **24** exhibited stronger anti-FAK activity (IC_50_ = 4.65 nM), which was stronger than that of **TAE-226**. Importantly, compound **24** also exhibited a potent capacity to decrease the growth of AsPC-1 and BxPC-3 cells with IC_50_ values 1.66 and 0.57 μM, respectively, which was lower than that of **TAE-226** (IC_50_ = 6.73 and 1.03 μM, respectively). Interestingly, compound **24** was not active against normal HPDE6-C7 cells (IC_50_ > 20 μM). In addition, compound **24** induced the apoptosis of AsPC-1 cells in dose and time-dependent manners and arrested AsPC-1 cells in a G2/M phase. The docking results indicated that compound **24** could well dock into the ATP-binding pocket of the FAK enzyme which is similar to lead compound **TAE-226** (Figure 10). It is worth noting that compound **24** forms three strong hydrogen bonds, which are produced by the N-1 atom in the pyrimidine core with amino acid Cys502, the carbonyl group in *N*-methylbenzamide with the amino acid Asp564, and the nitrogen atom in *N*-methylbenzamide with Arg505 and Gly506 through a water molecule. Notably, the morpholine substituent and the ethoxyl group of compound **24** could form strong contacts with the amino acid Gly505 and Glu506 through the polar-induced forces.

Small molecule inhibitors targeting FAK are usually designed to bind residues that surround the ATP-binding pocket of FAK, including **TAE-226**, **PF562271,** and **CEP-37440,** etc. *These inhibitors* are composed of pyrimidine core and *N*-methylbenzamide functional group. The co-crystal structure of the complex of FAK and TAE226 shows that the hydrophilic functional group on its C-2-aniline side chain is beneficial to strengthen the contact with the amino acids Gln438 and Gly505 in the FAK protein-binding pocket [63,85]. In fact, efforts in the structural modification of pyrimidine scaffolds have found several FAK inhibitors that are more effective than **TAE226**, such as compounds **18** and **24** (Figure 8 and Figure 10), which indicated that structural modifications such as the introduction of sulfonate or phosphonate groups at C-2-aniline side chain are beneficial to increase the binding affinity with FAK. Therefore, Ma’s group introduced the *N*-morpholine carboxamide functional group on the aniline side chain at the C-2 position of the pyrimidine core to design novel carboxamide-diphenylpyrimidines with strong hydrogen bond affinity to FAK in 2019 (Figure 11) [86].

Most carboxamide-diphenylpyrimidines displayed high inhibitory potency of FAK activity with IC_50_ values ranging from 2.58 to 75.8 nM. Among them, compound **25** exhibited equal anti-FAK activity (IC_50_ = 5.17 nM) to the lead compound **TAE-226** (IC_50_ = 5.5 nM). Significantly, compound **25** also displayed potent antiproliferation potency against AsPC-1, BxPC-3, and MCF-7/ADR cells (IC_50_ = 0.105, 0.09 and 0.59 μM, respectively), which were much lower than that of the lead compound **TAE-226** (IC_50_ = 6.73 μM, >10 and >10 μM, respectively). Compound **25** potently inhibited FAK phosphorylation in a dose-dependent manner and completely inhibited it at a concentration of 50 nM for 72 h in AsPC-1 cells, while the lead compound **TAE-226** completely inhibited FAK phosphorylation at a concentration of 100 nM. In addition, compound **25** also exhibited great antitumor efficacy in an AsPC-1 cancer xenograft mouse model. The results of molecule docking clearly suggested that compound **25** could tightly and similarly bind to the active site of FAK as the compound **TAE-226** (Figure 12). Notably, compound **25** formed a strong hydrogen bond between the oxygen atom of the amide group and amino acid Asp564, which is similar to that of **TAE-226** but also formed a new strong hydrogen bond through the carbonyl group with amino acid Gln438.

Recently, the discovery of irreversible covalent inhibitors targeting kinases has attracted great interest [87,88], which shows unique advantages, for example, irreversible covalent inhibitors could prolong the target retention time and obtain long-lasting inhibition without continuous administration, as well as reduce the development of drug-resistant mutations and better avoid resistance [89]. The crystal structure of FAK protein with **TAE-226** clearly indicates that Cys427 on the glycine-rich loop of FAK locates close to the ATP-binding site. Therefore, this cysteine residue could be used as the nucleophilic group for the design of covalent FAK inhibitors. Chen’s group successfully discovered novel FAK inhibitors by the introduction of an acrylamide moiety to the pyrimidine scaffold of **TAE-226** in 2018 (Figure 13) [90]. These compounds with appropriate linkers could ensure that acrylamide group can be in proximity to Cys427 of FAK protein and the 2,4-diphenylamine pyrimidine as a core scaffold to maintain the binding affinity with the FAK protein.

These compounds exhibited superior inhibitory potency against FAK with IC_50_ values ranging from 0.6 to 4.7 nM, which were lower than that of the lead compound **TAE-226**. Compound **30** bearing a squarate group displayed better inhibitory activity than compounds **32** and **33**. Compound **30** also exhibited excellent antiproliferation activity of squamous cell carcinoma (SCC) cells with a similar ED_50_ (IC_50_ = 1.73 μM) to that of **VS-4718** (IC_50_ = 1.49 μM). Besides, compound **30** significantly blocked Tyr397 phosphorylation and the autophosphorylation of FAK at low concentrations in SCC cells, which was comparable to **VS-4718**. The cocrystal structure of the FAK kinase domain with compound **30** shows that **30** makes a covalent bond with Cys 427 of the FAK kinase domain by means of its carbamate linker (Figure 14).

Molecular hybridization strategy is a common method of designing new drugs based on the combination of different biologically active parts to produce novel hybrids with higher affinity and effectiveness [91]. Before 2019, there was no report on 2,4-diarylaminopyrimidine dithiocarbamate hybrid compounds. The combination of 2,4-diarylaminopyrimidine and dithiocarbamate into one compound exhibited the potential to enhance its anti-FAK activity and antiproliferative effects [92,93]. Yin’s group designed a class of 2,4-diarylaminopyrimidines as novel FAK inhibitors by incorporating the dithiocarbamate moiety in 2019 (Figure 15) [94]. It is found that the different structural units of R have a great influence on FAK and cell inhibitory activity. Among of these compounds, compound **34** exhibited the highest inhibitory potency of FAK (inhibition rate = 73% at 1 μM). Compound **34** was selected as a new starting point for further optimization and retained the *N*,*N*-dimethylaminoethyl group. However, the introduction of a strong electron-withdrawing group, such as trifluoromethyl (compound **35**), could significantly increase the FAK-inhibitory activity. Compound **35** exhibited more potent inhibitory activity against FAK (IC_50_ = 0.07 nM) compared to **TAE-226** and excellent antiproliferative effects against HCT-116, PC-3, U87-MG, and MCF-7 cell lines with IC_50_ values of 1, 30, 60, and 20 nM, respectively, which were significantly lower than that of the lead compound **TAE-226** (IC_50_ = 290, 1680, 1660, and 560 nM, respectively). Moreover, compound **35** significantly induced apoptosis by arresting cell cycle in G2/M phase in both MCF-7 and HCT-116 cell lines.

In recent years, with the development of polypharmacology and a deep understanding of tumor pathogenesis, multi-targeting antitumor agents have been considered as an important strategy to overcome drug resistance and improve therapeutic effects. Based on a fragment-based drug design strategy, Chen’s group discovered a new class of diphenylpyrimidine derivatives as potent FAK and EGFR^T790M^ dual inhibitors in 2020 (Figure 16) [95]. Epidermal growth factor receptor (EGFR) tyrosine kinase has been proved to be a valuable clinical target for anticancer therapy, especially for NSCLC [96,97]. However, the T790M mutation (threonine to methionine) within the ATP site of the EGFR frequently occurs in NSCLC patients (about 60%), and these patients develop resistance to drugs that target EGFR, which leads to the failure of EGFR-targeted therapy [98,99]. In addition, novel EGFR^T790M^ inhibitors such as **CO-1686** (Phase III) and **AZD-9291** consist of the core scaffold of pyrimidine. Therefore, they designed and synthesized the first novel dual FAK and EGFR^T790M^ inhibitors using a fragment-based drug design strategy based on the lead compound **TAE-226** and **Rociletinib**. Fortunately, most of them exhibited great inhibitory potency of FAK and EGFR ^T790M^.

Among these compounds, compound **41** not only exhibited high inhibitory activity of FAK kinase (IC_50_ = 1.03 nM) and EGFR ^T790M^ (IC_50_ = 3.89 nM) but also displayed excellent antiproliferative activity against AsPC-1, BxPC-3, Panc-1 three FAK-overexpressing pancreatic cancer cells (IC_50_ = 0.909, 0.761, 1.218 μM, respectively) and two drug-resistant cancer cell lines MCF-7/ADR and H1975 cells (IC_50_ = 0.909, 0.761, 1.218 μM, respectively), which were significantly lower than that of the lead compound **TAE-226**. In addition, compound **41** also showed the efficiency in the in vivo assessment conducted in a FAK-driven human AsPC-1 cell xenograft mouse model.

Chen’s group reported another new class of irreversible covalent FAK inhibitors based on compound **30** in 2020 (Figure 17) [100]. Among them, compound **45** exhibited more effective and selective inhibition potency of FAK with an IC_50_ value of 0.6 nM than that of **TAE-226**. Compound **45** potently suppressed the proliferative activity of three human glioblastoma cell lines U-87 MG, A172, and U251 with IC_50_ values of 7.2, 0.89, and 1.2 μM, respectively, which were lower than that of the lead compound **TAE-226** (IC_50_ = 2.9, 8.3 and 6.3 μM, respectively). Compound **45** could significantly suppress the autophosphorylation of FAK in U-87 MG cells at low concentrations and block the phosphorylation of FAK-Tyr397 in a dose-dependent manner. In addition, compound **45** also significantly reduced the rate of cell migration and blocked U-87 cells in the G2/M phase.

Positron emission tomography (PET) is an advanced clinical imaging technology in the field of nuclear medicine. In recent years, many groups discovered novel small molecule PET-imaging agents through ^18^F labeling strategies based on anticancer agents with dual biological properties. Zhang’s group first reported F-18-labelled 2,4-diaminopyrimidines as FAK inhibitors and tumor-imaging agents (Figure 18) [101]. These compounds displayed superior inhibition potency against FAK with IC_50_ values ranging from 5.0 nM to 205.1 nM. Interestingly, compounds **52** and **53** which were labeled by ^18^F exhibited respective IC_50_ values of 5.0 nM and 21.6 nM. In addition, compound **52** exhibited promising biodistribution data in sarcoma S180-bearing mice, with tumor/blood, tumor/muscle, and tumor/bone ratios of 1.17, 2.99, and 2.19, respectively.

Zhang’s group reported novel 2,4-diaminopyrimidine derivatives FAK as tumor radiotracers in 2021 (Figure 19) [102]. Among them, compounds **54**, **55**, and **58** exhibited potently inhibitory activity against FAK with IC_50_ values of 3.0, 0.6, and 3.2 nM, respectively. Replacement of the chlorine atom in the pyrimidine ring with other groups assisted in penetrating the protein pocket and maintaining a low IC_50_ level. Furthermore, compound **58** with a FAK titer was radiologically labeled [18F]. Compound **58** presented high radiochemical purity (>98%), high in vitro biostability, and good tumor uptake values in S180-bearing mice, with a tumor/muscle ratio of 2.08 and a tumor/bone ratio of 2.48 at 30 min of injection.

### 3.3. FAK Inhibitors of Pyrimidine Heterocyclic Compounds

2,4-Diaminopyrimidine is the main scaffold of FAK inhibitors, which exhibited the inhibitory activity against FAK and cancer cells. Therefore, many groups adopted the scaffold hopping strategy, replacing the core pyrimidine scaffold with pyrimidine heterocyclics, such as 7*H*-pyrrolo[2,3-*d*]pyrimidine-2,4-diamine scaffold and thieno[3,2-*d*]pyrimidine scaffold to design novel FAK inhibitors.

7*H*-pyrrolo[2,3-*d*]pyrimidine derivative **59** (Figure 20) is a novel FAK inhibitor reported by Cheng’s group in 2019 [103]. They focused on the replacement of the 2,4-diaminopyrimidine core with the pyrimidine heterocyclic scaffold. They assumed it is feasible to introduce additional interactions with Glu500 by simply adding a hydrogen donor in this region, which would lead to the hinge binding part containing three donor-acceptor interactions, while the pyrimidine contains two cases. Besides, they introduced various substituents of R_1_ with hydrogen bond acceptors to form hydrogen bond interactions with Asp564 of the DFG motif. The introduction of dimethyl phosphine oxide moiety at R_1_ was favorable to increase activity against FAK. In addition, they further explored SAR at R_2_ with the dimethyl phosphine oxide substituted phenyl group at R_1_ position. A further modification is to introduce a substituted benzene ring or a substituted pyrazole ring at R_2_ to improve the efficacy and adjust the properties, and finally, focus on the R_3_ position of the phenyl group in position R_2_ on the pyrimidine. Since the para-substitution carrier R_3_ points to the solvent region, the introduction of hydrophilic fragments at the R_3_ position can help improve activity. A class of fragment-containing derivatives with different hydrophilic properties were discovered, and most of these compounds showed strong inhibitory potency against FAK. They also designed and synthesized compounds **61, 62, 63,** and **64** containing thieno[2,3-*d*]pyrimidine or thieno[3,2-*d*]pyrimidine scaffold based on representative compounds **59** and **60** as FAK inhibitors.

Among them, compounds **59** and **60** exhibited great inhibitory potency of FAK with IC_50_ values of 5.4 and 23.1 nM, respectively. Compared with compounds **59** and **60**, the inhibitory activity of thieno-pyrimidine analogs **61**, **62**, **63**, and **64** against FAK was reduced by more than five times. These results indicate that the hydrogen bond between 7*H*-pyrrolo[2,3-*d*]pyrimidine and Glu500 helps to obtain high efficiency. In addition, compound **59** displayed inhibition activity against A549 cells with an IC_50_ value of 3.2 µM, and also induced A549 cells apoptosis and suppressed its migration in a dose-dependent manner. Moreover, compound **59** exhibited good metabolic stability in mouse, rat, and human liver microsomes. The results of the docking study indicated compound **59** was anchored to the hinge region via the canonical donor-acceptor-donor hydrogen-bonding motif between the nitrogen molecules on the 7*H*-pyrrolo[2,3-*d*]pyrimidine-2,4-diamine moiety and the backbone of residues Glu500 and Cys502 (Figure 21). The hydrophobic interaction by hydrophobic side chains of the Ala 452 and Leu 553 helps increase further stabilization. The dimethyl phosphine oxide moiety interacts with Asp564 of the DFG motif through hydrogen bonding.

Thieno [3,2-*d*]pyrimidine derivative **67** is a potent FAK inhibitor reported by Cheng’s group in **2020** [104] (Figure 22). Based on a cyclization strategy, Cheng’s group used 2,7-disubstituted-thieno[3,2-*d*]pyrimidine scaffold to mimic the bioactive conformation of the well-known diaminopyrimidine scaffold of FAK inhibitors. First, they focused on the R_1_ moiety by fixing the R_2_ moiety as a phosphonate group. In R_1_ moiety, different fragments containing hydrogen bond acceptors were introduced to form hydrogen bond interactions with Asp564 of the DFG motif. Compound **65** contains a 2-methoxy group at the R_1_ position, which exhibited the strongest inhibitory potency on FAK (IC_50_ = 134 nM). Then, Cheng’s group performed SAR detection at R_2_. The introduction of the *N*-substituted formamide group (compound **66**, IC_50_ = 38.8 nM) produced twice the effectiveness of compound **65**. Based on compound **66**, the substituents of the benzene ring were changed to further explore the SAR. The introduction of F at 3-position of benzene ring maintained high efficacy against FAK (IC_50_ = 28.8 nM). In addition, they also explored the influence of the changes on the solubilizing groups at the R_6_ position.

The optimized piperidine-4-yl-substituted compound **67** has a high inhibitory activity against FAK (IC_50_ = 28.2 nM). Compound **67** also exhibited excellent antiproliferation activity against U-87MG, A549, and MDA-MB-231 cells (IC_50_ = 0.16, 0.27, and 0.19 µM, respectively), which were significantly lower than that of the lead compound **TAE-226** (IC_50_ = 1.67, 1.16, and 4.06 µM, respectively). In addition, compound **67** could significantly induce cell apoptosis, arrest cell cycle at the G0/G1 phase, and suppress migration in MDA-MB-231 cells. The results of the docking study elucidate the possible binding modes of compound **67** and provide a structural basis for the further design of FAK inhibitors (Figure 23).

Gray’s group reported a new class of FAK inhibitors with a tricyclic benzopyrimidodiazepinone core in 2021 [105]. Previous studies suggested that the tricyclic benzopyrimidodiazepinone scaffold is a privileged scaffold for the design of kinase inhibitors, for example, **BJG-01-181** (compound **72**) was identified as a FAK inhibitor (IC_50_ = 62.2 nM) in the biochemical kinase assay. Therefore, they explored the structure–activity relationships of this novel scaffold to improve the inhibitory potency of FAK (Figure 24). In addition, the *N*-methylated compounds usually have better pharmacokinetic (PK) properties than the secondary amides [106], so they designed *N*-methylated tricyclic benzopyrimidodiazepinone core to provide compounds including compounds **74**, **75**, and **76** with enhanced FAK inhibitory potency. Based on compound **73**, they further explored SAR of the substituent group on benzopyrimidodiazepinone core including compounds **77** and **78**.

Compound **73**, an *N*-methyl matched-pair with compound **72**, was synthesized and exhibited a three-fold improvement against FAK with an IC_50_ value of 20.2 nM. A variety of modifications provided inhibitors **74**, **75**, and **76** with IC_50_ values in the 30–60 nM range. *N*-ethylation at any position of the central ring retained nearly identical inhibitory potency of FAK as compound **73**. In addition, compound **73** exhibited potent activity in 3D-culture breast and gastric cancer models, and favorable pharmacokinetic properties in mice. The results of the docking study suggested that compound **73** revealed two hydrogen-bonding interactions with FAK protein, which might explain the potent inhibitory of compound **73** (Figure 25).

### 3.4. FAK Inhibitors with a Triazine Scaffold

Triazine ring is often reported as an important scaffold for anticancer drugs [107,108]. For example, furazil and dioxadet, which contain different amino groups at positions 2, 4, or 6, have been reported as anticancer drugs. Because the chlorine atom of the triazine nucleus can easily be substituted successively with nucleophilic groups, resulting in a variety of substitutions, many groups develop new FAK inhibitors with a triazine scaffold.

Pyrrolo[2,1-*f*][1,2,4]triazine **82** is a potent inhibitor of dual FAK and JAK2 developed by Zificsak’s group in 2012 [109]. Pyrrolo[2,1-*f*][1,2,4]triazine usually was designed as a core scaffold of ALK inhibitors, and Zificsak’s group screened **79** showing potent inhibitory activity against JAK2 (IC_50_ = 0.2 nM) and good potency of FAK (IC_50_ = 17 nM). In addition, bioavailable ALK inhibitor **80** showed potent FAK activity but modest inhibition of JAK2. Therefore, Zificsak’s group designed a new class of pyrrolo[2,1-*f*][1,2,4]triazines as dual inhibitors of FAK and JAK2 based on the compounds **79** and **80** (Figure 26). Compound **81** showed strongest inhibitory effect on JAK (IC_50_ = 2.0 nM) and FAK (IC_50_ = 14 nM). Compound 81 was further optimized by introducing a chlorine atom into the pyrrolo [2,1-*f*][1,2,4]triazine ring to obtain compound **82**. Progress has been made in FAK and JAK2 dual inhibitors, which showed potent inhibitory activity against both FAK and JAK2. In addition, compound **82** potently inhibited *p*-STAT3 and *p*-FAK phosphorylation in CWR22 tumor mice xenografts.

Diarylamino-1,3,5-triazine derivative **87** was identified as a novel FAK inhibitor reported by Chen’s group in 2013 [110] (Figure 27). Chen’s group first introduced a 3,4,5-trimethoxyanilino group and a methanesulfonamide phenyl group on the 1,3,5-triazine ring to obtain compound **85** with poor activity against FAK (IC_50_ = 41.9 µM). Next, they removed the chlorine atom of the triazine ring based on compound **85** to give compound **86**, which showed approximately eight-fold inhibitory activity against FAK (IC_50_ = 5.1 µM). Substituting an amide moiety (compound **87**) for the complex sulfonamide group (compound **86**) could significantly increase the inhibitory activity 13 times against FAK (IC_50_ = 0.4 µM). Chen’s group also explored the influence of the changes of the 3,4,5-trimethoxyanilino group on FAK activity (compounds **88**, **89**, **90**, and **91**).

Among of these compounds, compound **88** exhibited high inhibitory activity against both FAK (IC_50_ = 0.31 µM) and HUVEC cells growth (IC_50_ = 1.5 µM). In addition, the crystal structure of compound **88** in the FAK protein was resolved (PDB ID: 4brx), which indicated compound **88** interacting with the FAK kinase domain is highly similar to that of the lead compound **TAE-226** (Figure 28).

Chen’s group reported novel imidazo[1,2-*a*][1,3,5]triazines as FAK inhibitors in 2015 [111]. Based on the studies of triazine FAK inhibitor **87**, they designed novel compounds by the fusing imidazole ring with the triazine to give an imidazo[1,2-*a*][1,3,5]triazine (Figure 29). Imidazo[1,2-*a*][1,3,5]triazine **92** exhibited stronger binding affinity to FAK (2-fold) than that of compound **87**, showing specific contributions of imidazo[1,2-*a*][1,3,5]triazine. SAR was further explored and Chen’s group designed compounds **94** and **95** with a CH_2_NH_2_ group or CH_2_NHAlloc group in the para position of the simple phenyl ring, which exhibited more potent inhibitory activity of FAK (IC_50_ = 120 and 50 nM, respectively) than that of compound **87**.

Compared to **TAE-226**, compound **95** significantly inhibited the autophosphorylation of FAK Y397 in HCT-116, MDA-MB-231, U-87MG, and PC-3 cells. In addition, compound **95** exhibited excellent antiproliferation activity against HCT-116, and PC-3 cells (IC_50_ = 0.52, and 0.29 μM, respectively). Compound **95** also potently inhibited the cell-matrix adhesion, migration, and invasion of U87-MG cells in a dose-dependent manner. The results of the molecular docking studies suggested that compound **95** is very similar as observed for compound **87** (Figure 30). However, imidazo[1,2-*a*] ring could make additional interactions, most notably with Met499, CH_2_NH group forms a hydrogen bond interaction with Ile-428, and the CO of the alloc group in compound **95** also could produce an additional interaction with Glu506, which might explain the increased binding affinity of compound **95** to FAK.

Although the interaction mode of **TAE-226** and compound **88** is very similar, compound **88** exhibited lower in vitro potency of FAK about 45 times than that of **TAE-226.** Chen’s group proposed that one major reason from X-ray structure is the missing chlorine atom in 1,3,5-triazine compounds, which in **TAE-226** makes van der Waals interactions with Met499 of FAK kinase, therefore, Chen’s group developed 1,2,4-triazine compounds as a novel scaffold for FAK inhibitors which can contain a chlorine atom at position 6 of 1,2,4-triazine core in 2017 (Figure 31) [112].

Among these compounds, compound **98** containing a 2-*OCH_3_*-4-morpholino group showed the best anti-FAK activity with an IC_50_ value of 230 nM and excellent antiproliferation activity against HCT-116 cells (IC_50_ = 0.19 µM). The results of the docking studies suggested that the van der Waals interactions between the chlorine atom and Met499 of FAK protein might not be important to the binding of compound **98** with FAK protein, suggesting that the weaker hydrogen bonds between the amino 1,2,4-triazine and the hinge region of FAK may play an important role in the inhibitory activity of compound **98** (Figure 32).

### 3.5. Other Types of FAK Inhibitors

Many groups also reported five-membered heterocyclic compounds containing nitrogen as FAK inhibitor (Figure 33), including 1,3,4-thiadiazole derivatives **102** and **103** [113,114], 1,3,4-oxadiazole derivatives **104** and **105** [115,116], 2-styryl-5-nitroimidazole derivatives **106** [117], 1,2,4-triazole derivative **107** [118]. Most of these compounds exhibited inhibitory activity and antiproliferative activity at the micromolar levels.

Zhu’s group reported a new class of novel 1,3,4-thiadiazoles as FAK inhibitors. The most active compound **102** inhibited the activity of FAK and HEPG2 cells with an EC_50_ value of 10.79 and 10.28 μM, respectively (Figure 33). Furthermore, Zhu’s group designed and synthesized 1,3,4-thiadiazol-2-amide derivatives with more potent inhibitory potency of FAK and cancer cells growth compared to **102** [114]. The most active compound **103** exhibited better inhibitory activity against FAK with an IC_50_ value of 5.32 μM and inhibited proliferation of MCF-7 and B16-F10 cells with IC_50_ values of 0.45 and 0.31 μM, respectively [113]. The results of docking studies suggested that compounds **102** and **103** could be well embedded in the ATP-binding pocket of FAK through a hydrogen bond, π–σ interaction, and π–cation interaction (Figure 34).

Zhu’s group designed a new class of 1,3,4-oxadiazole-2(3*H*)-thiones as FAK inhibitors by replacing the 1,3,4-thiadiazole core of compounds **102** and **103** with further modifications. The most active compound **104** exhibited high inhibitory potency of FAK at submicromolar level (IC_50_ = 0.78 μM) and good inhibition of HepG2 and SM1116 cell growth (IC_50_ = 5.78 and 47.15 μM, respectively) [115]. Altintop’s group also reported a new class of oxadiazole-based derivatives as FAK inhibitors. The most active compound **105** exhibited significantly the inhibitory activity of FAK (Phospho-Tyr397) with an IC_50_ value of 19.5 μM in C6 cells [116]. Compound **105** also showed potent anticancer activity against C6 cells (IC_50_ = 4.63 μM) and caused higher caspase-3 activation and morphological changes.

In 2014, Zhu’s group also designed novel 2-styryl-5-nitroimidazole derivatives as FAK inhibitors with antiproliferative activity [117]. Compound **106** exhibited potently inhibitory activity against FAK (IC_50_ = 0.45 μM), and A549 and HeLa cells growth with IC_50_ values of 0.45, 3.11 μM, respectively. In addition, compound **106** induced HeLa cells apoptosis in a dose-dependent manner. Compound **106** could decrease the phosphorylation of PI3K, Akt, JNK, and STAT3, resulting in apoptosis induction and cell cycle arrest. The results of molecular docking revealed that compound **106** could well fit within the ATP-binding site of FAK. In 2021, Hayallah’s group reported a new class of 1,2,4-triazole derivatives as FAK inhibitors with potential anticancer activity [118]. The most active compound **107** inhibited the activity of FAK with an IC_50_ value of 18.10 nM and inhibited HepG2 and Hep3B cells growth with IC_50_ values of 3.78 and 4.83 μM, respectively.

Miki’s group reported 1,5-dihydropyrazolo[4,3-*c*][1,2]benzothiazines as novel FAK inhibitors by high-throughput screening (HTS) (Figure 35). The HTS of the internal compound library of this research group gave more than ten hit compounds of different chemical classes, and further modification gives compounds **109** and **110.** Compound **109** and **110** inhibited the FAK activity with IC_50_ values 0.64 and 4.2 μM, respectively [119].

## 4. The Discovery of FAK-Targeting PROTACs and Their Therapeutic Significance

The proteolytic targeting chimera (PROTAC) is a new chemical knockout technique for post-translational proteins. PROTAC is a heterogeneous bifunctional small molecule containing two recognition parts: one specifically binds to E3 ubiquitin ligase, and the other specifically binds to the target protein. The PROTAC molecule can drive the E3 ubiquitin ligase to the target protein, which leads to the ubiquitination of the target protein, which leads to proteasome-mediated degradation (Figure 36) [120,121].

Crew’s group reported potent FAK selective degraders in 2018 [122]. In this work, they designed FAK degraders by conjugating a FAK ligand derived from **VS-6063** (**defactinib**) through slight modifications with a peptide-based VHL ligand or thalidomide through ether linkers (Figure 37). All of these FAK PROTACs exhibited good inhibitory potency against FAK and could significantly induce the degradation of FAK at low concentrations. Among the FAK-PROTACs, compound **111** exhibited potent activity, which showed an IC_50_ value of 6.5 nM, DC_50_ value of 3 nM, and Dmax over 99%. Compound **111** displayed better selectivity to FAK than **VS-6063**. In addition, compound **111** displayed significant effects in suppressing the FAK downstream signaling pathways, which could highly induce the degradation of FAK in a dose-dependent manner with only 34% total FAK remaining at 10 nM and 5% at 50 nM, which outperformed **VS-6063** (**defactinib**). In addition, compound **111** significantly reduces *p*-paxillin levels by as much as 85–90% at 50 nM, *p*-Akt suppression of 93% at 1 μM, and is more effective than **VS-6063**. Compound **111** also significantly inhibited the migration and invasion of MDA-MB-23 cells.

Ettmayer’s group reported a new class of FAK-PROTACs represented by compounds **113** and **114** in 2019, by tethering a previously identified FAK inhibitor compound **112** through an ether linker with a peptide-based VHL ligand and pomalidomide, respectively [123] (Figure 38). Both of FAK-PROTACs (compound **113** and **114**) showed comparable degradation potencies and efficacies with mean pDC_50_ values of 7.45 and 7.08 nM, respectively. The CRBN-based PROTAC **113** potently degraded FAK (DC_50_ = 27 nM) with a Dmax of 95% at the chosen experimental conditions, while the VHL-based PROTAC **114** only achieved a Dmax of 80% and was less potent (DC_50_ = 243 nM) in the A549 cells. However, although the potent depletion of FAK, both FAK-PROTACs (compound **113** and **114**) did not exhibit antiproliferation activity against the tested cell lines in either short or long-term assays, which indicated that scaffolding function of FAK may not be required for the proliferation of the tested cell lines.

Rao’s group reported a new class of FAK-PROTACs in 2020, by tethering a previously identified FAK inhibitors VS-6062 or VS-6063 through different ether linker with a CRBN E3 ligand pomalidomide [124] (Figure 39). They noticed that shorter diethylene or triethylene glycol linkers exhibited higher degradation activity of FAK. Among them, compound **115** caused rapid and significant degradation of FAK with DC_50_ values at the scale of picomolar in TM3, PA1, MDA-MB-436, LNCaP, and Ramos cells (DC_50_ = 310, 80, 330, 370, and 40 pM, respectively). Compound **115** showed a significant inhibitory effect on the autophosphorylation of FAK (pFAK^tyr397^) below 1 nM, but **VS-6062** showed an inhibitory effect on pFAK^tyr397^ only at 3 μM. However, the efficient knockdown of FAK by compound **115** did not more severely affect the proliferation of the tested cell lines.

Gray’s group reported a new class of FAK-PROTACs by tethering a previously identified FAK inhibitor **VS-4718** through an ether linker with a CRBN E3 ligand pomalidomide in a 2018 patent. All of them could effectively produce degradation of FAK with a Dmax of 85–100% at 100 nM, and compound **120**, **121**, **122**, and **123** only achieved it at 10 nM [125] (Figure 40).

## 5. Concluding Remarks and Prospects

FAK is an intracellular non-receptor tyrosine kinase that promotes tumor cell growth by controlling cell adhesion, migration, proliferation, and survival. Therefore, targeting FAK is considered to be a promising cancer therapy with small molecules.

At present, a variety of FAK inhibitors have been proven to exhibit good antitumor activity. A comprehensive analysis of the mode of action of various FAK inhibitors shows that most of the inhibitor cores are composed of five-membered or six-membered nitrogen-containing heterocycles, which can be selectively inserted into the ATP-binding pocket of FAK and interact with the kinase hinge. The nitrogen atoms on the heterocycle could form key hydrogen bonds with the amino acid residues of FAK, and the heterocycle itself can form hydrophobic interactions with the surrounding residues. At the same time, the interaction between the branch chain of the core and FAK can further enhance the activity of the inhibitor, and the modification of the branch chain can also improve the drug ability of the inhibitor. FAK inhibitors have special effects on the FAK protein through the active groups in the structure, inhibiting the phosphorylation of phosphorylation sites (such as Tyr 397, Tyr 576, and Tyr 577) in the FAK structure, and inhibiting the overexpression of FAK, thereby exerting antitumor effect. So far, there is no approved FAK inhibitor in the market. Therefore, there is an urgent need to develop new FAK inhibitors with higher antitumor activity to exert stronger antitumor effects. Combining more anticancer therapies to treat human cancer is a frequently used strategy. The joint research of FAK inhibitors and other anticancer drugs will also be in the direction of FAK inhibitor development. PROTAC technology has also been applied to the development of FAK drugs. Several potential FAK PROTACs have been reported, which opens a new window for FAK-targeted therapy.

## Figures and Tables

**Figure 1 molecules-26-04250-f001:**
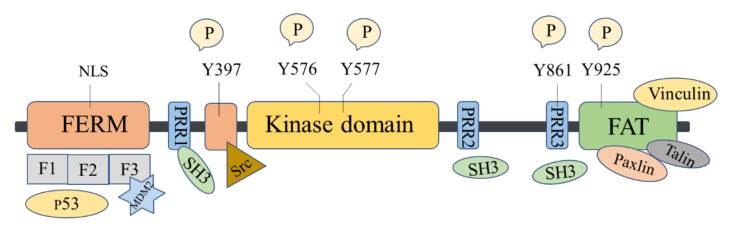
Schematic diagram of the molecular structure of FAK protein.

**Figure 2 molecules-26-04250-f002:**
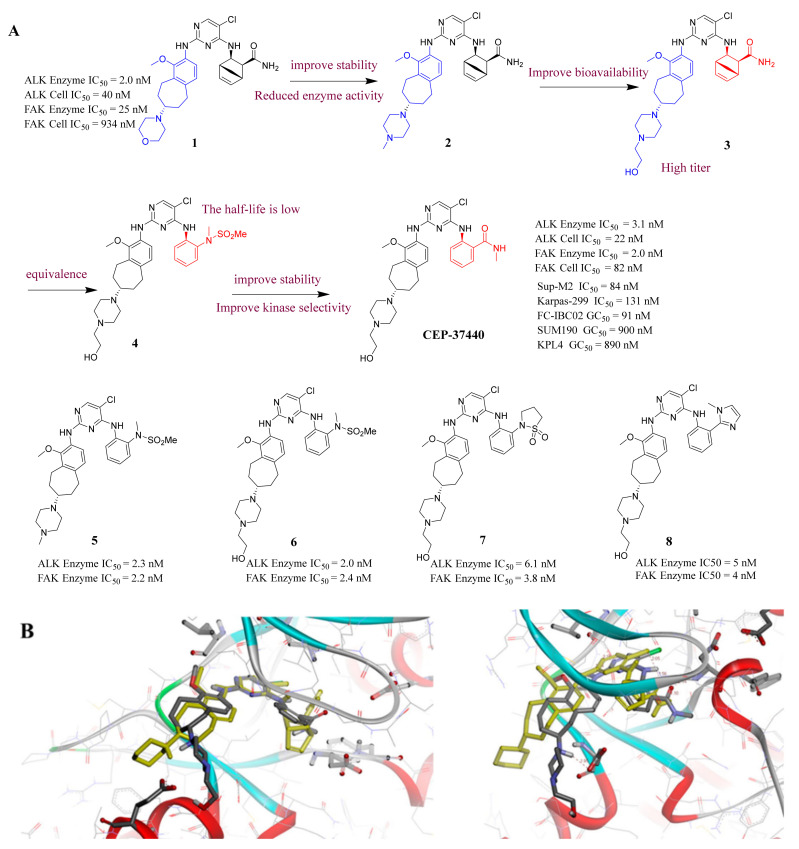
(**A**) The discovery of compound **CEP-37440****.** (**B**) Glide docking of **CEP37440** (gray) and compound **1** (yellow): left panel, ALK (PDB code 3LZT); right panel, FAK (PDB code 3BZ3).

**Figure 3 molecules-26-04250-f003:**
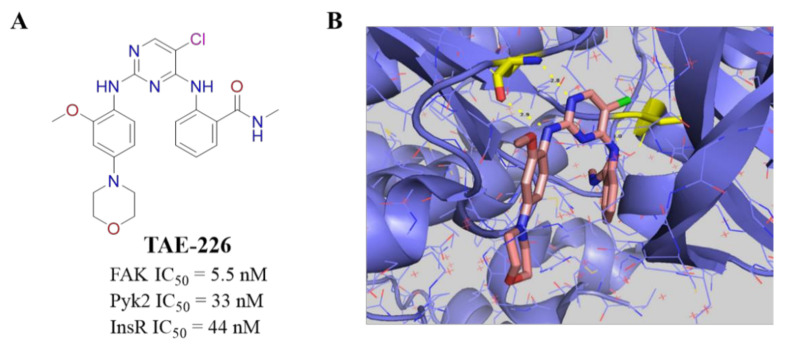
(**A**) Chemical structure of **TAE-226**. (**B**) Crystal structure of **TAE-226** complexed with the kinase domain of FAK (PDB ID: 2JKK). Hydrogen bonds are shown in yellow dash lines. The residues that can form hydrogen bonds with **TAE-226** are shown in yellow.

**Figure 4 molecules-26-04250-f004:**
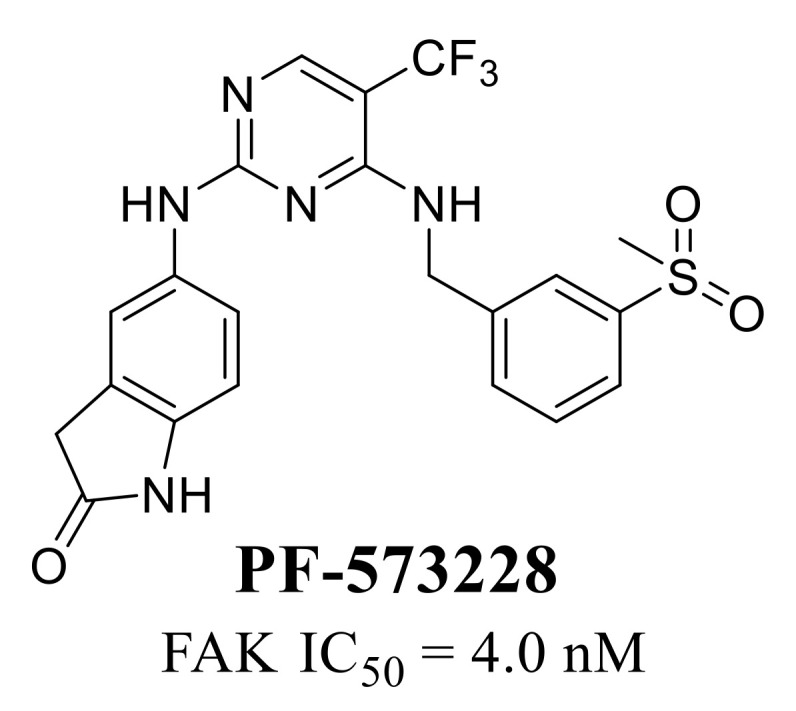
Chemical structure of **PF-573228**.

**Figure 5 molecules-26-04250-f005:**
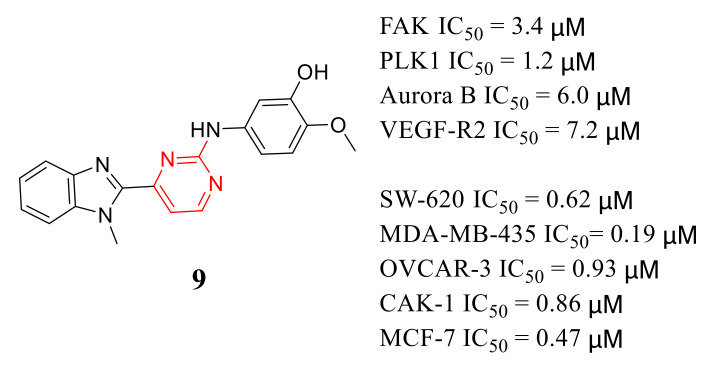
Chemical structure and inhibition activity of compound **9**.

**Figure 6 molecules-26-04250-f006:**
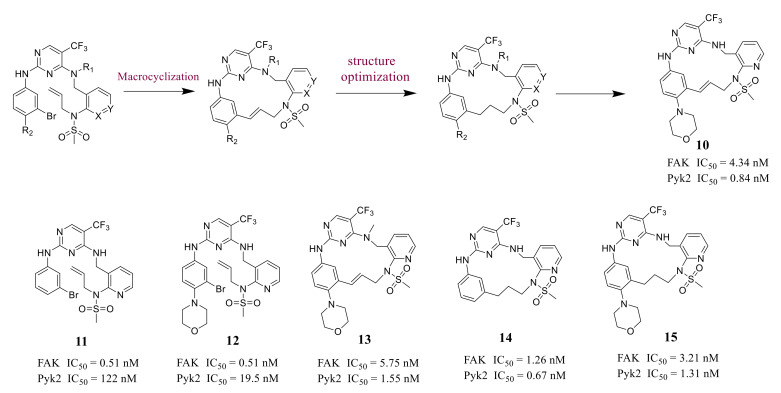
Macrocyclic double-target inhibitors of FAK and Pyk2 were obtained by cyclization.

**Figure 7 molecules-26-04250-f007:**
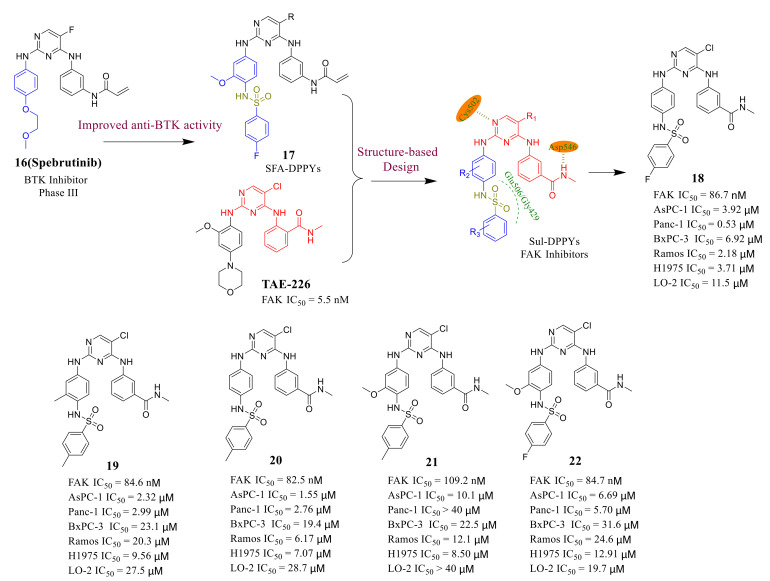
Discovery of diphenylpyrimidine derivatives with sulfonamide functional groups as FAK inhibitors.

**Figure 8 molecules-26-04250-f008:**
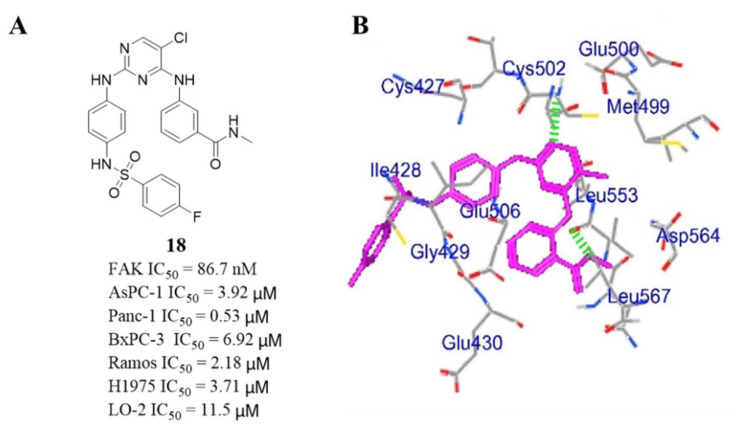
(**A**) Chemical structure of compound **18.** (**B**) Predicted binding pose of compound **18** in FAK enzyme (PDB code: 2JKK).

**Figure 9 molecules-26-04250-f009:**
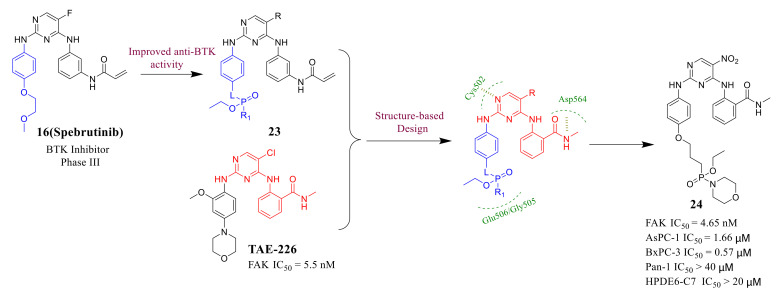
The fragment-based drug design strategy of compound **24**.

**Figure 10 molecules-26-04250-f010:**
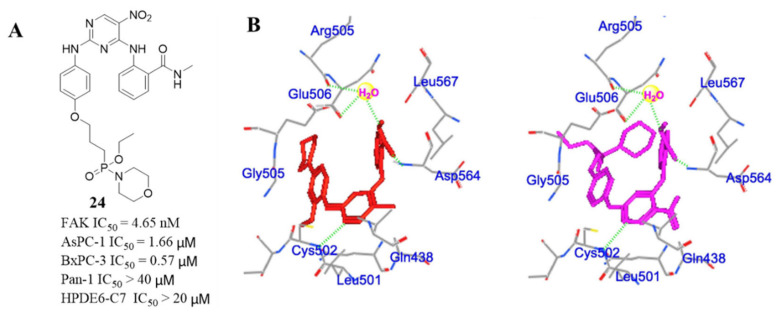
(**A**) Chemical structure of compound **24****.** (**B**) Predicted binding models of **TAE-226** (red) and compound **24** (violet) in FAK enzyme (PDB code: 2JKK).

**Figure 11 molecules-26-04250-f011:**
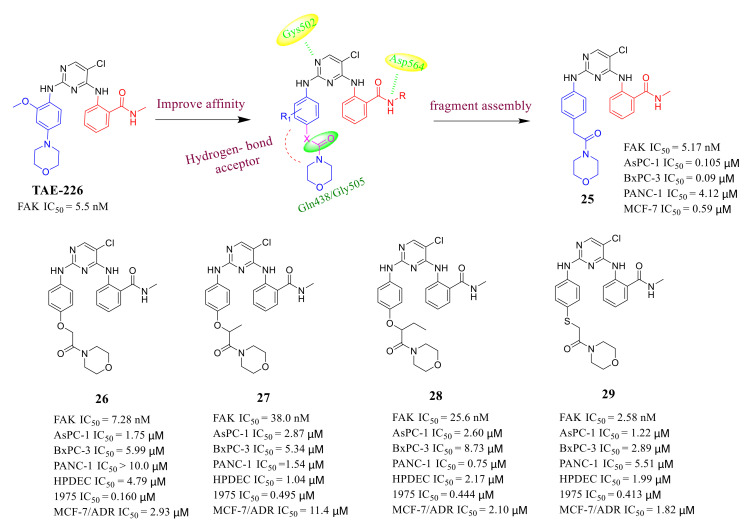
Compound **25** was obtained by structural modification.

**Figure 12 molecules-26-04250-f012:**
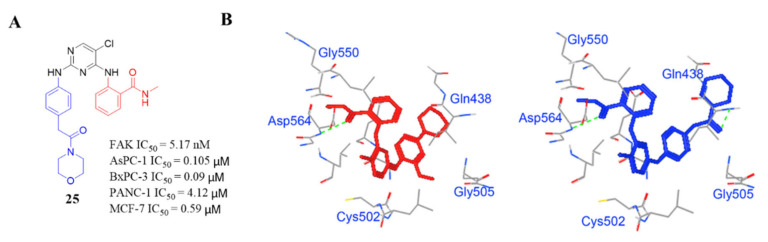
(**A**) Chemical structure of compound **25**. (**B**) Predicted binding models of **TAE-226** (red) and compound **25** (blue) in FAK enzyme (PDB code: 2JKK).

**Figure 13 molecules-26-04250-f013:**
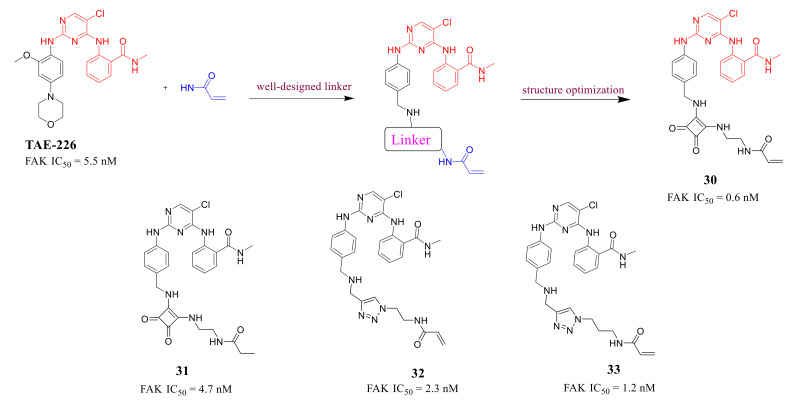
The design strategy of first irreversible covalent FAK inhibitors.

**Figure 14 molecules-26-04250-f014:**
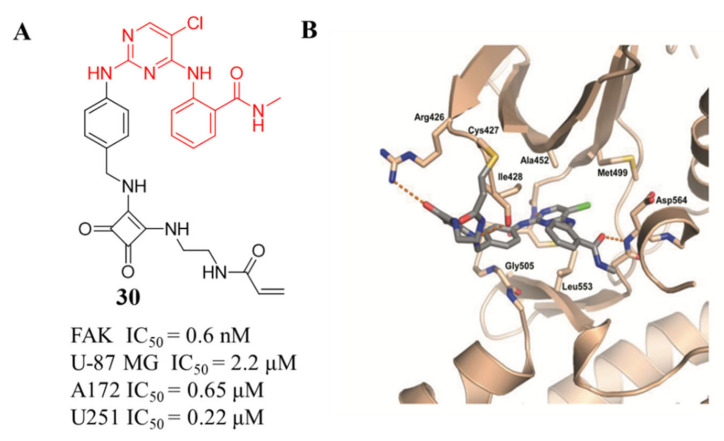
(**A**) Chemical structure of compound **30****.** (**B**) Crystal structure of compound **30** (PDB ID: 6GCX) linked the kinase domain of FAK.

**Figure 15 molecules-26-04250-f015:**
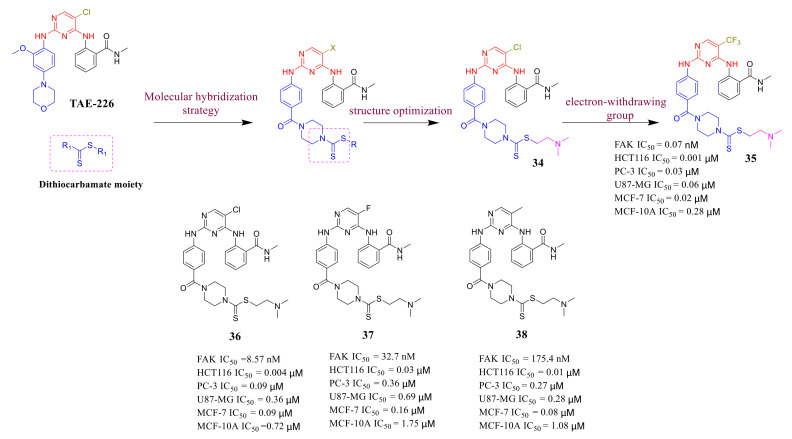
The design strategy of 2,4-diarylaminopyrimidine derivatives by incorporating the dithiocarbamate moiety.

**Figure 16 molecules-26-04250-f016:**
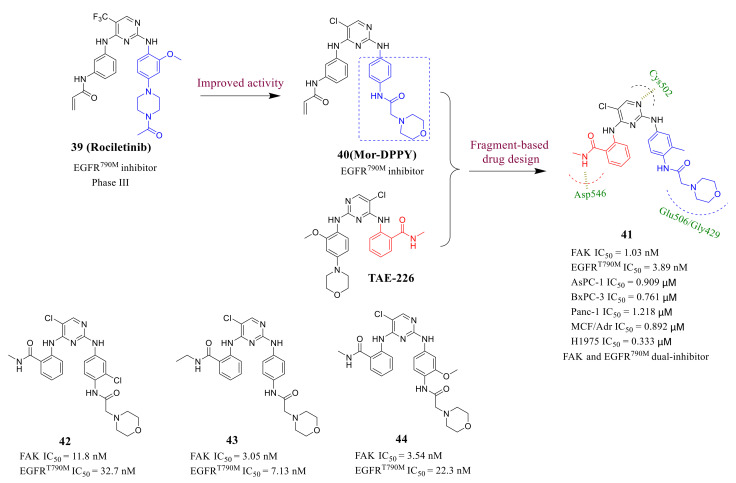
The designed strategy of dual FAK and EGFR^T790M^ inhibitors.

**Figure 17 molecules-26-04250-f017:**
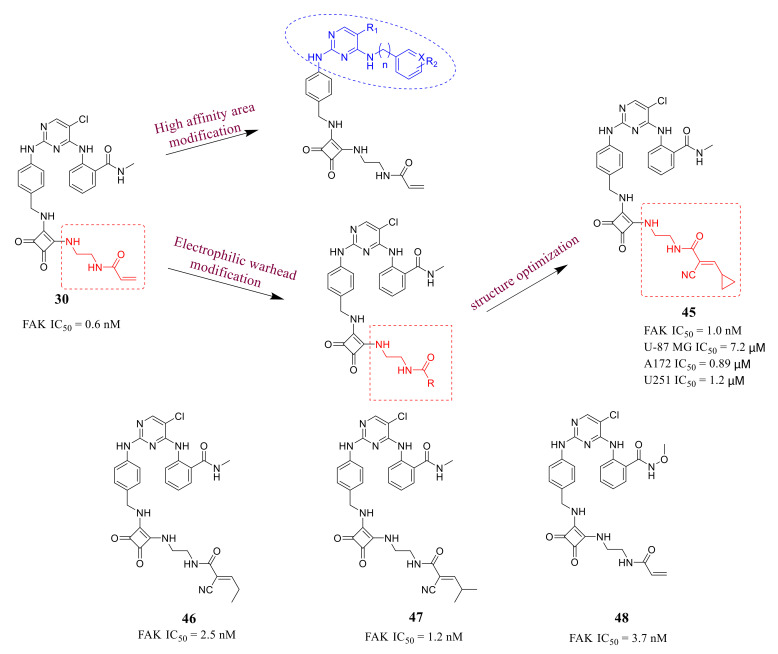
Discovery of reversible valence FAK inhibitors. Compound **30** and compound **45** came from the same research group.

**Figure 18 molecules-26-04250-f018:**
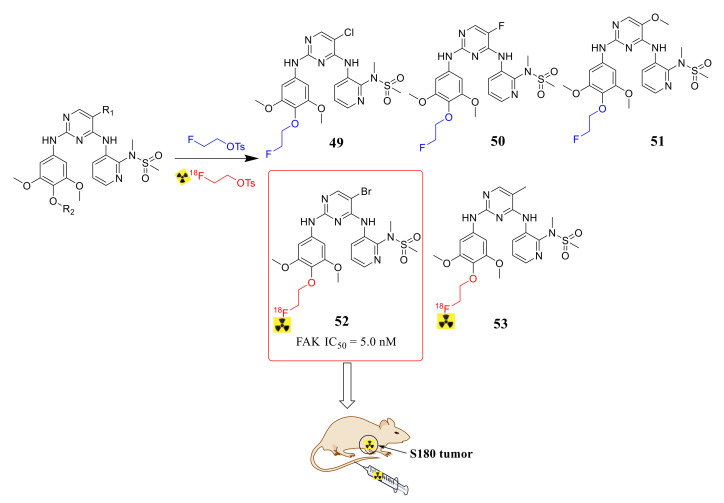
Design and synthesis of F-18-labelled 2, 4-diaminopyrimidine-type FAK-targeted inhibitors as potential tumor-imaging agents.

**Figure 19 molecules-26-04250-f019:**
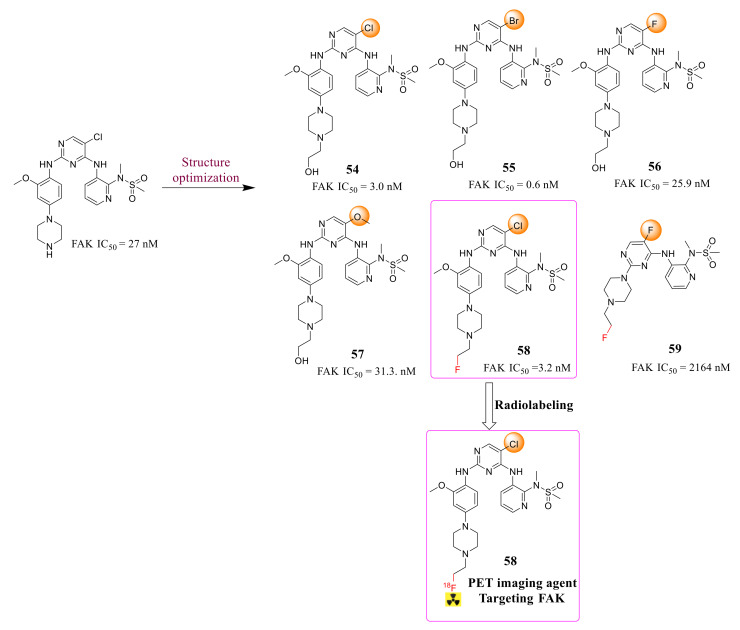
Design of 2,4-diaminopyrimidine derivatives targeting focal adhesion kinase as tumor radiotracers.

**Figure 20 molecules-26-04250-f020:**
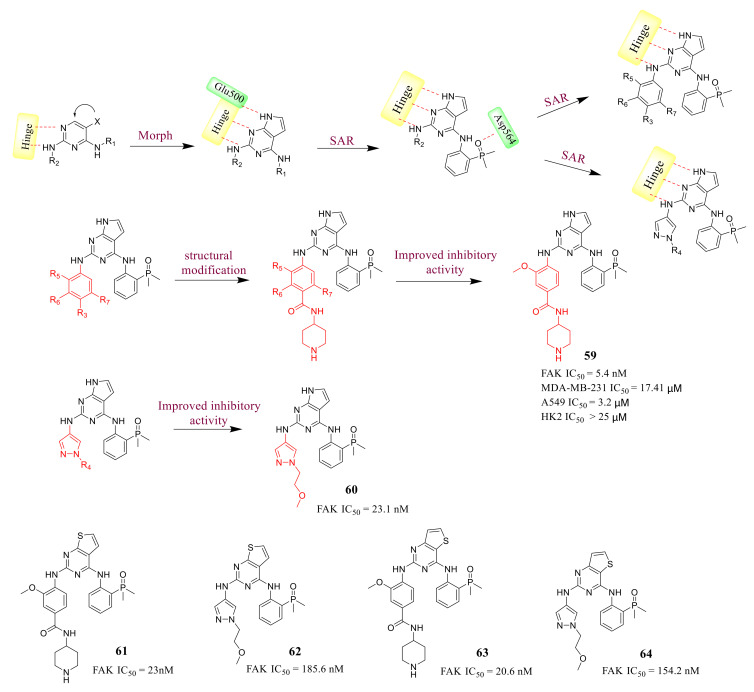
Compounds **59** and **61** replaces 2,4-diaminopyrimidine scaffolds by pyrrolo-pyrimidine 2,4-diamine and thieno-pyrimidine 2,4-diamine scaffolds.

**Figure 21 molecules-26-04250-f021:**
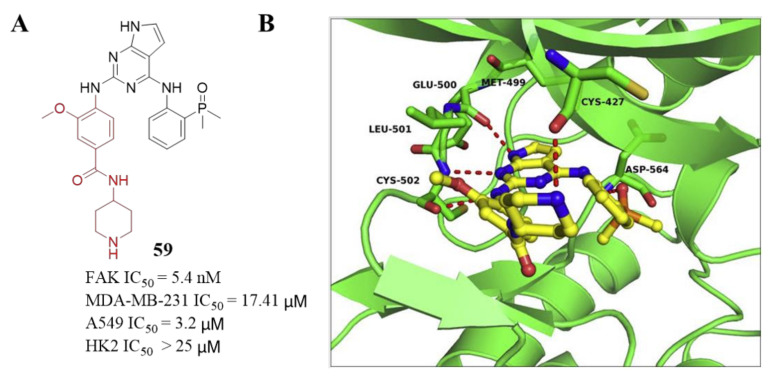
(**A**) Chemical structure of compound **59****.** (**B**) Predicted binding mode of compound **59** in the ATP-binding site of FAK (PDB ID: 2JKK).

**Figure 22 molecules-26-04250-f022:**
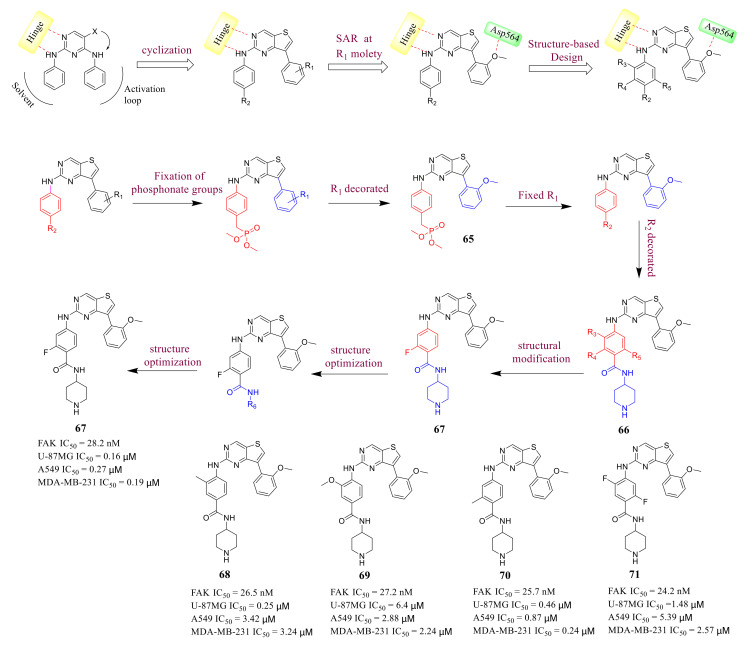
Design and synthesis of thieno [3, 2-*d*]pyrimidine derivatives as potent FAK inhibitors.

**Figure 23 molecules-26-04250-f023:**
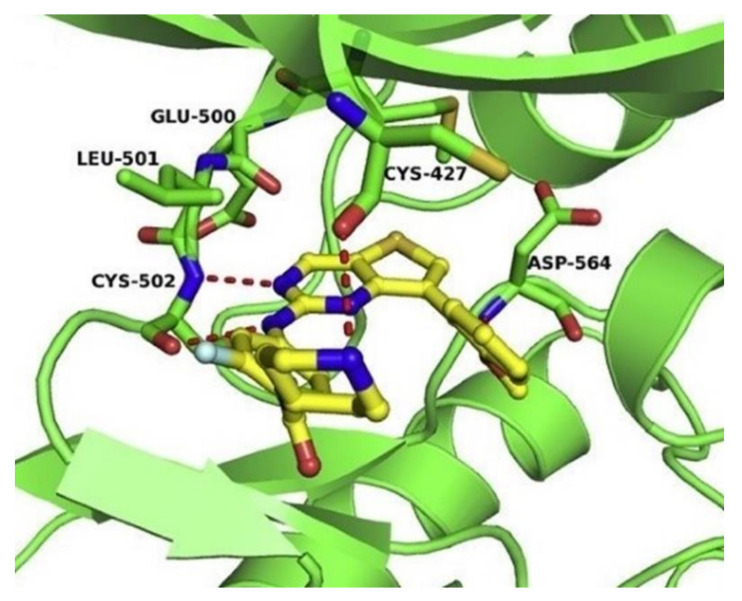
The predicted docked pose of compound **67** (yellow sticks) in the FAK active site (PDB: 2JKK). Detailed interactions with the protein residues. Each dashed red line represents hydrogen bonds.

**Figure 24 molecules-26-04250-f024:**
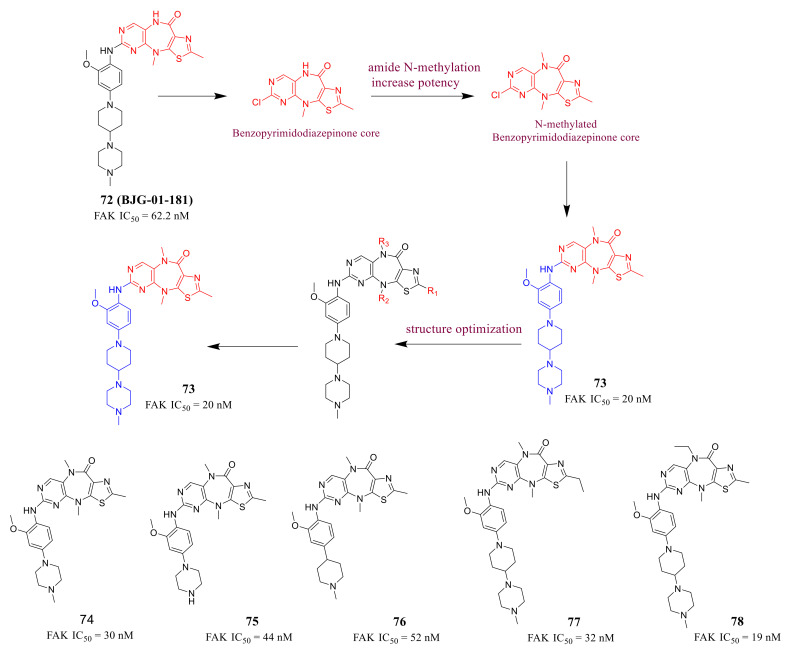
Discovery of a pyrimidothiazolodiazepinone as a potent and selective focal adhesion kinase (FAK) inhibitor.

**Figure 25 molecules-26-04250-f025:**
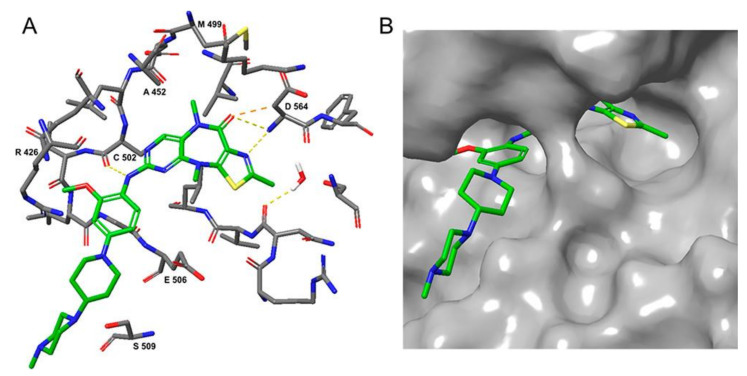
Docking of compound **73** with FAK (PDB 6I8Z). (**A**) Key interactions of compound **73** with active-site residues, notably the critical hydrogen-bond of the thiazole nitrogen with Asp564. (**B**) FAK surface map shows the U-shaped active site, with the 2-methyl group extending into a hydrophobic pocket.

**Figure 26 molecules-26-04250-f026:**
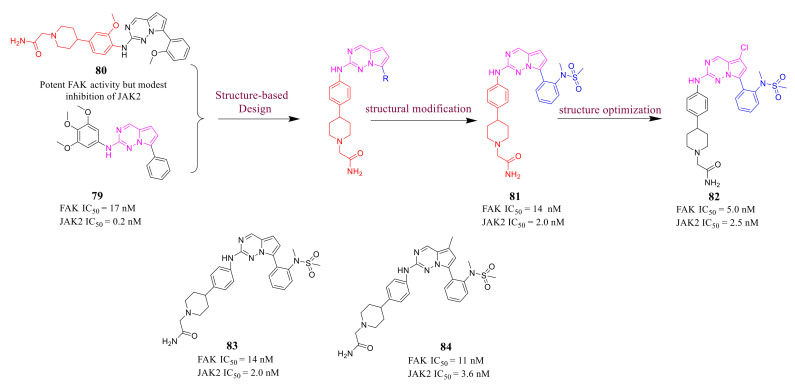
Optimization of a novel kinase inhibitor scaffold for the dual inhibition of JAK2 and FAK kinases.

**Figure 27 molecules-26-04250-f027:**
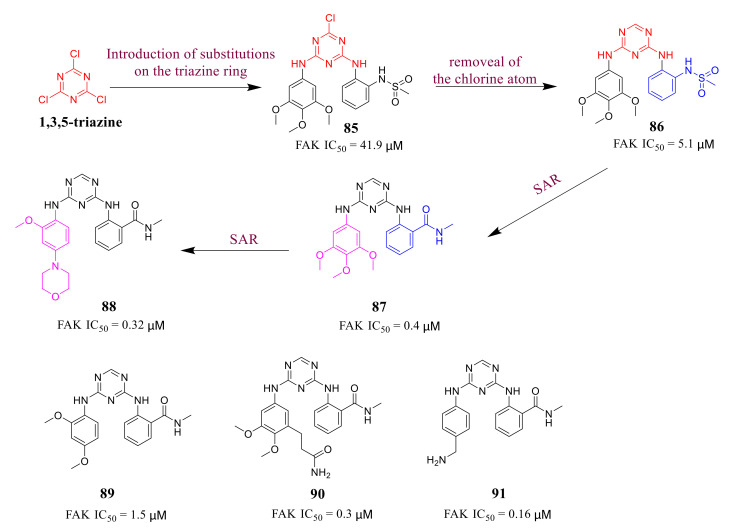
Designs and synthesis of diarylamino 1,3,5-triazine derivatives as FAK inhibitors.

**Figure 28 molecules-26-04250-f028:**
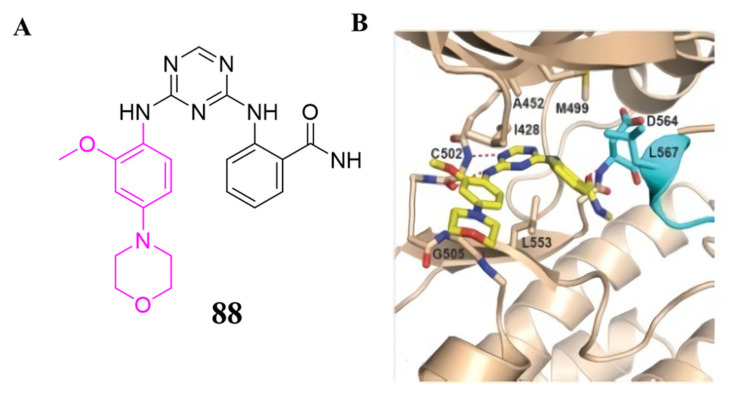
(**A**)**.** Chemical structure of compound **88**. (**B**)**.** Compound **88** is shown to bound to the active site of the FAK kinase (beige ribbon with activation looping cyan, PDB ID: 4brx). Key side chains and the inhibitors (yellow) are shown in stick representation.

**Figure 29 molecules-26-04250-f029:**
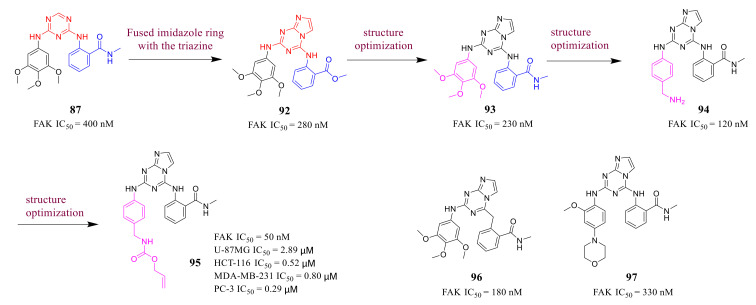
Designs and synthesis of imidazo [1,2-*a*][1,3,5]triazine cyclic FAK inhibitors.

**Figure 30 molecules-26-04250-f030:**
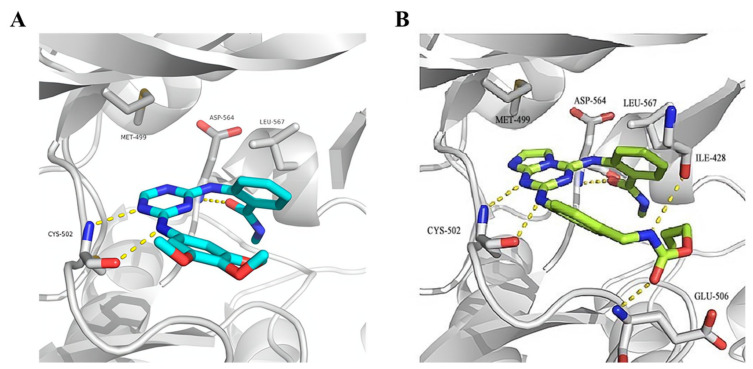
Docking of compound **87** (**A**) and **95** (**B**) into the ATP-binding pocket of FAK (PDB ID: 4c7t).

**Figure 31 molecules-26-04250-f031:**
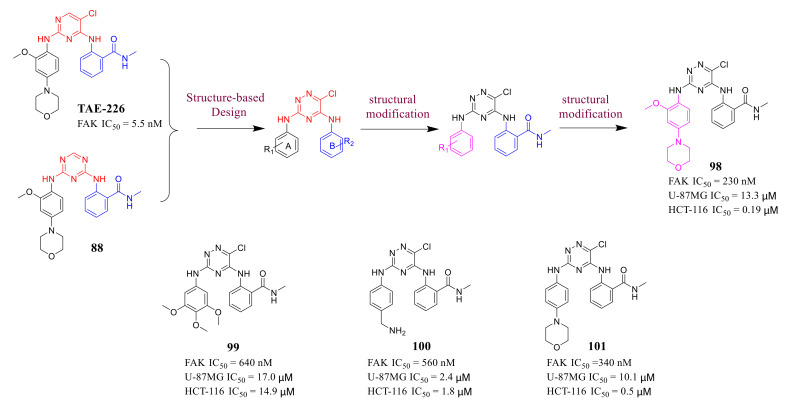
Design and synthesis of 1,2,4-triazine derivatives as FAK inhibitors.

**Figure 32 molecules-26-04250-f032:**
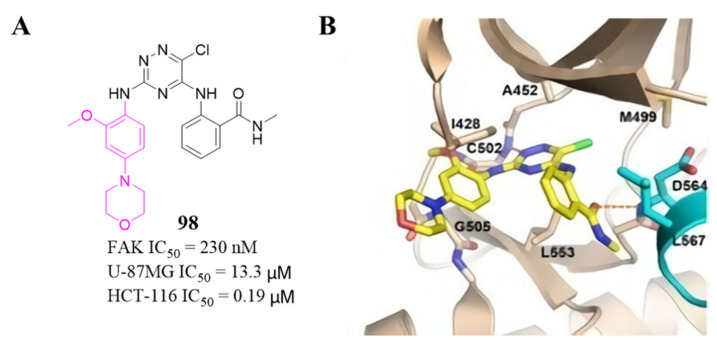
(**A**) Chemical structure of compound **98**. (**B**) compound **98** is shown bound to the active site of the FAK kinase (beige ribbon with activation looping cyan, PDB ID: 4brx). Key side chains and the inhibitors (yellow) are shown in stick representation.

**Figure 33 molecules-26-04250-f033:**
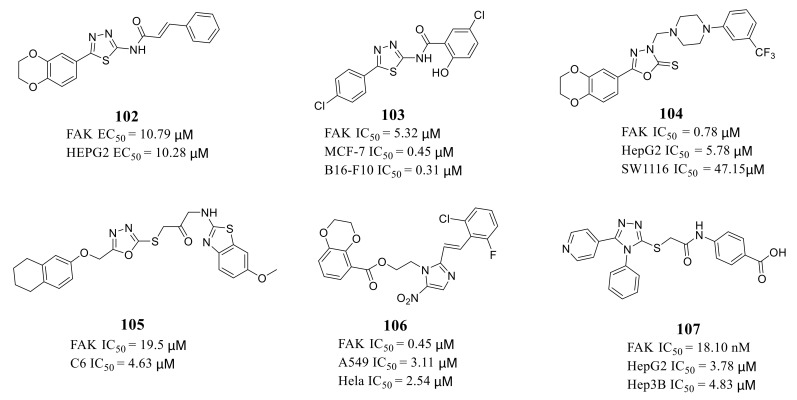
Nitrogen-containing 5-membered heterocyclic FAK inhibitors.

**Figure 34 molecules-26-04250-f034:**
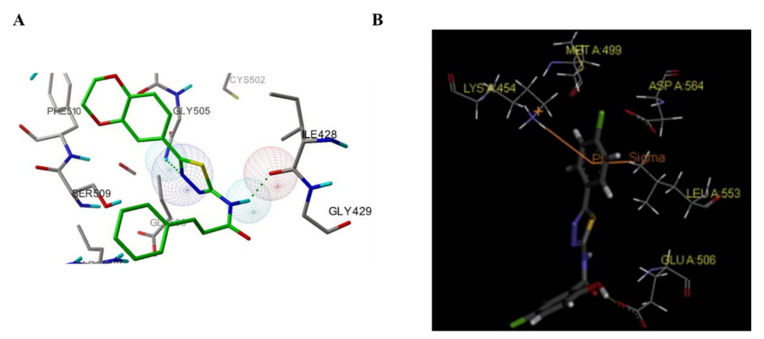
Molecular docking modeling of compound **102** (**A**) and compound **103** (**B**) with FAK.

**Figure 35 molecules-26-04250-f035:**
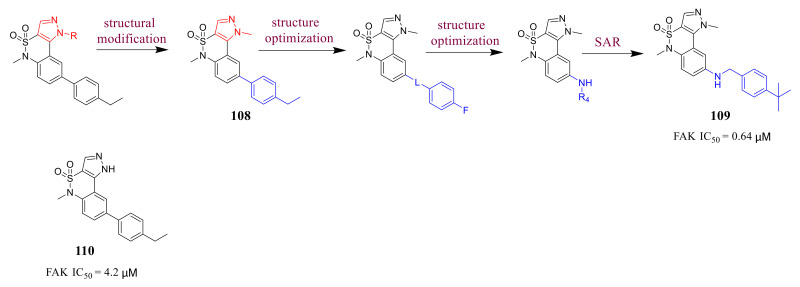
Design and synthesis of an allosteric inhibitor **110** targeting FAK.

**Figure 36 molecules-26-04250-f036:**
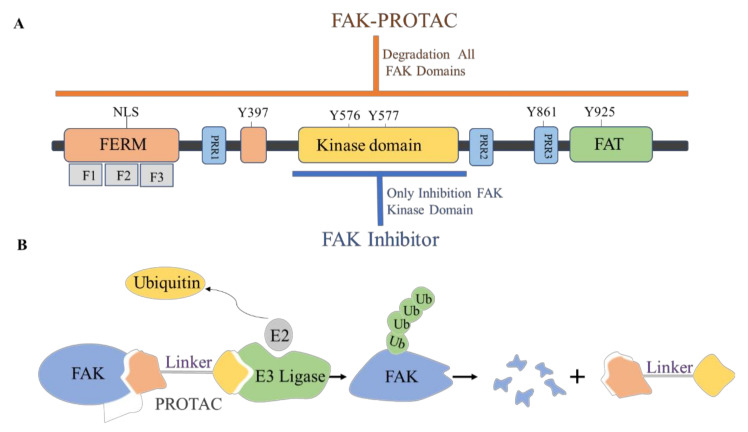
(**A**) FAK protein schematic. FAK-PROTACs can act on both enzymatic and nonenzymatic functions of FAK, while FAK inhibitor only acts on the enzymatic function of FAK. (**B**) Schematic depiction of the PROTAC strategy.

**Figure 37 molecules-26-04250-f037:**
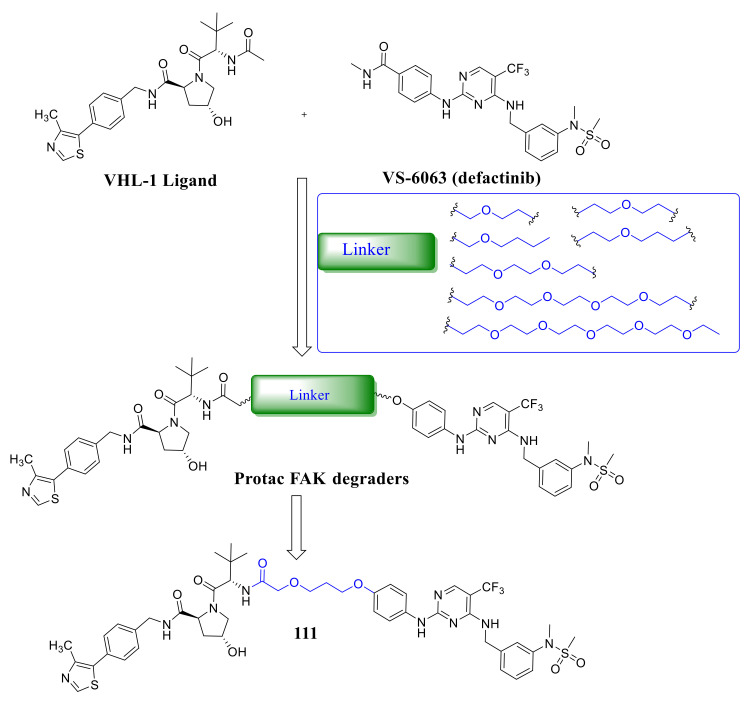
Bifunctional pyrimidine 111 as a proteolytic targeting chimera of focal adhesion kinase (FAK).

**Figure 38 molecules-26-04250-f038:**
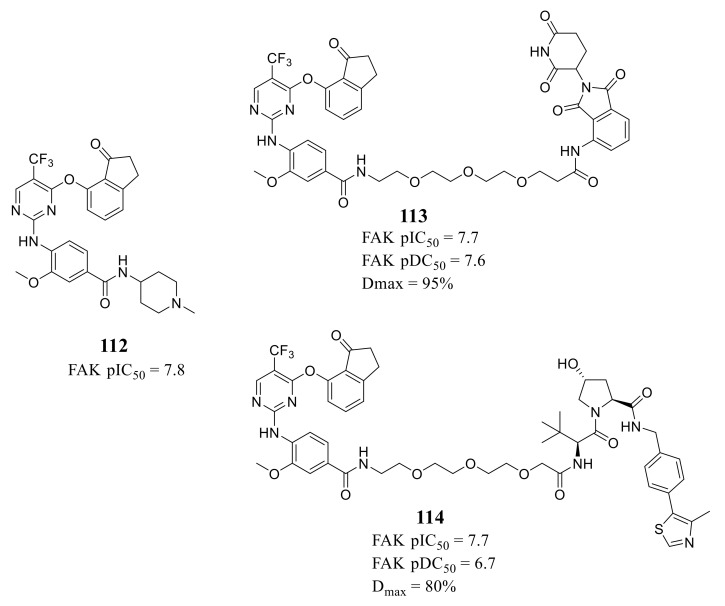
Compounds **113** and **114** as a proteolytic targeting chimera of FAK.

**Figure 39 molecules-26-04250-f039:**
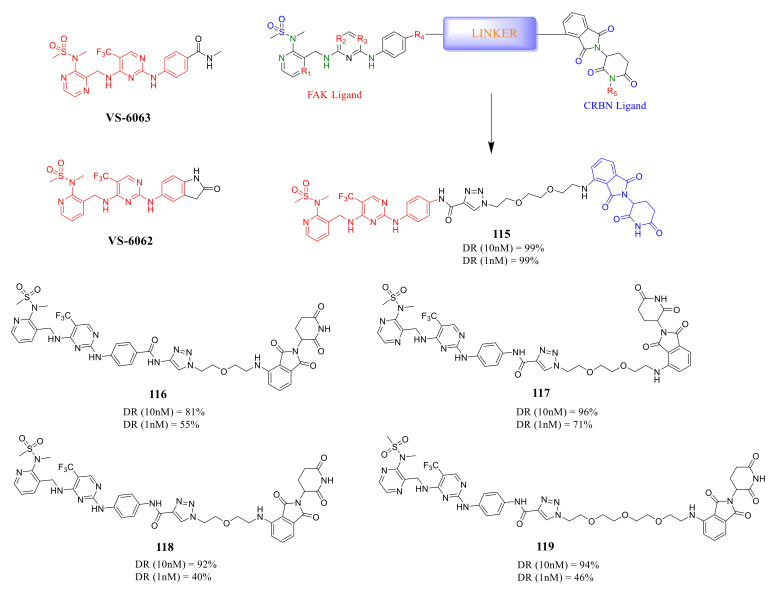
Compound **115** as a proteolytic targeting chimera of focal adhesion kinase.

**Figure 40 molecules-26-04250-f040:**
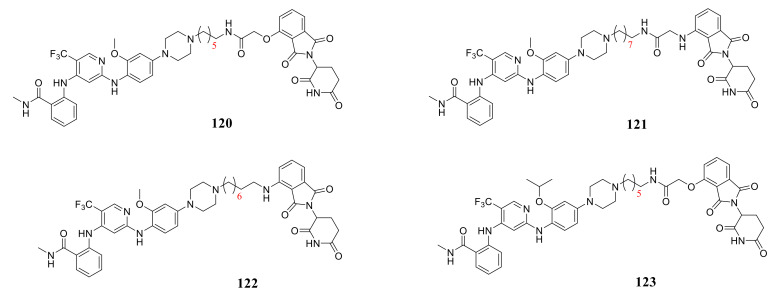
The proteolytic targeting chimera of focal adhesion kinase.

**Table 1 molecules-26-04250-t001:** All reported FAK inhibitors in clinical trials.

Inhibitor	Structure	Target (IC_50_)	Company	Clinical Trial	Status	Identifier
**GSK-2256098**	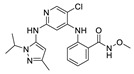	FAK IC_50_ = 0.4 nM	Glaxo Smith Kline	phase I	completed completed completed	NCT01138033 NCT00996671 NCT02551653
**VS-6063** **(PF-04554878)** **(Defactinib)**	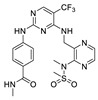	FAK IC_50_ = 0.6 nM Pyk2 IC_50_ = 0.6 nM	Verastem	phase II	completed completed completed	NCT01943292 NCT00787033 NCT01951690
**VS-6062** **(PF-00562271)**	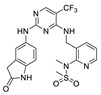	FAK IC_50_ = 1.5 nM Pyk2 IC_50_ = 14 nM	Verastem	phase I	completed	NCT00666926
**VS-4718** **(PND-1186)**	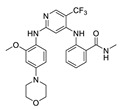	FAK IC_50_ = 1.5 nM	Verastem	phase I	terminated withdrawn	NCT01849744 NCT02215629
**CEP-37440**	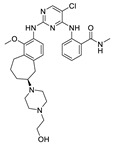	FAK IC_50_ = 2 nM ALK IC_50_ = 3.1 nM	Teva Branded	phase I	completed	NCT01922752
**BI-853520** **(IN-10018)**	Unknown	FAK	Boehringer Ingelheim	phase I	completed completed	NCT01335269 NCT01905111

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
