# Peer review of "Drug Discovery Targeting Focal Adhesion Kinase (FAK) as a Promising Cancer Therapy"

_molecules, 2021, doi:10.3390/molecules26144250_

Round 1

Reviewer 1 Report

The manuscript concerns the extensive information available on the inhibition of non-receptor tyrosine kinase activity by small molecules. The particular emphasis of this manuscript is on the chemical structures of lead compounds and their binding affinities. Inhibition data for approximately 120 derivatives are presented, which is an impressive amount of information. This information could be really useful for many researchers in the field of structure-based drug design of FAK inhibitors. However, the article will benefit from editing for English language since there are many sentences that take an effort to understand or do not make sense (as example, lines 263-264: „Ma‘s group design novel phosphamide by designed novel phosponamide. Because these compounds are phosphonamides. in 2017 [87]“.

Grammar is an issue throughout the manuscript.

Several comments that need author's attention:

Line 130. I did not find the IC50 values of compounds VS-6063 and VS-6062 in reference 43, please cite the original work.

Line184. The term „potent potency“ is not used, it should be „high or great potency“.

Line 192. „FAK inhibitor 1 named by TAE-226“. However, compound 1 structure showed in figure2 is not the same as for TAE-226. Please check.

Line 267 and line 304. Authors should check compound numeration presented in Figures 9 and 11. Different compound structures have the same numbers (compounds 25-28).

Line 349. Cys527 should be changed to Cys427.

Line 498. Please check the legend of the Figure 22. It is unclear what is written in the brackets [3, 2-d]

Line 525. Two structures of compound 73 are shown in Figure 24. Presumably the structure of this compound on the right of the figure is unnecessary.

Line 587. The term „triazine 92 exhibited stronger potency affinity“ is not used . It is better to use „Compound exhibited stronger binding affinity“.

Lead compounds are incorrectly named as „led compounds“ throughout the text.

Author Response

Answers to Reviewer #1:

1. Grammar is an issue throughout the manuscript.

Response: Thanks for your valuable comments. We feel sorry for our carelessness and have corrected the grammar mistakes in the revised manuscript.

2. Line 130. I did not find the IC50 values of compounds VS-6063 and VS-6062 in reference 43, please cite the original work.

Response: Thanks for your valuable comments. We feel sorry for our carelessness and

have deleted the IC50 values of compounds VS-6063 for the sake of more accurate description in the revised manuscript.

3. Line184. The term “potent potency” is not used, it should be “high or great potency”.

Response: Thanks for your valuable comments. We feel sorry for our carelessness and have corrected it according to your valuable comments in the revised manuscript.

 4. Line 192. “FAK inhibitor 1 named by TAE-226”. However, compound 1 structure showed in figure2 is not the same as for TAE-226. Please check.

Response: Thanks for your valuable comments. We feel sorry for our carelessness and have corrected the mistake in the revised manuscript.

5. Line 267 and line 304. Authors should check compound numeration presented in Figures 9 and 11. Different compound structures have the same numbers (compounds 25-28).

Response: Thanks for your valuable comments. We feel sorry for our carelessness and have corrected the mistake in the revised manuscript.

6. Line 349. Cys527 should be changed to Cys427.

Response: Thanks for your valuable comments. We feel sorry for our carelessness and have corrected the mistake in the revised manuscript.

7. Line 498. Please check the legend of the Figure 22. It is unclear what is written in the brackets [3, 2-d]

Response: Thanks for your valuable comments. We feel sorry for our carelessness and have corrected the mistake in the revised manuscript.

8. Line 525. Two structures of compound 73 are shown in Figure 24. Presumably the structure of this compound on the right of the figure is unnecessary.

Response: Thanks for your valuable comments. We feel sorry for our carelessness and have corrected it according to your valuable comments in the revised manuscript.

9. Line 587. The term “triazine 92 exhibited stronger potency affinity“”is not used . It is better to use “Compound exhibited stronger binding affinity”.

Response: Thanks for your valuable comments. We feel sorry for our carelessness and have corrected it according to your valuable comments in the revised manuscript.

10.Lead compounds are incorrectly named as “led compounds”throughout the text.

Response: Thanks for your valuable comments. We feel sorry for our carelessness and have corrected the mistake in the revised manuscript.

Reviewer 2 Report

Authors present their review on FAK inhibitors. Topic is interesting to the readers of Molecules, as well article is reasonably well set up, although there are several instances where article must be improved:

- language should eb checked and in some cases, verbs are missing.

- consistency in writing should be checked  e.g. writing HeLa cells and Hela cells

- Frequency of the use of the phrase Chen’s group .... should be reduced.

- in several instances H, N, O, should be used in italic

There are also some exact examples of corrections needed

Page 2, line 70: "Another new FAK, CEP-37440" should be corrected to "Another new FAK inhibitor, CEP-37440"

Page 3, line 111: "targeting FAK is emerging" as there are 6 ongoing clinical trials, targeting FAK can not be emerging.

Page 3, line 116: Table 1: size of chemical structures should be unified (e.g. all aromatic rings should be of identical size)

Page 9: Figure 7: correct "BTK inhibitors" to "BTK inhibitor"

Page 10, line 263: Somebody forgot about the final revision of this paragraphs, which currently reads "Ma’s group design novel phosphamide by designed novel phosphonamide. Because these compounds are phosphonamides . in 2017 [87]"

Page 13, line 316: listing docking energies is irrelevant here, not mentioning the accuracy (four digital places) - (Energy score: −16.5776 kJ mol−1)

Page 16, Figure 15: Presenting numerical values should eb unified for helping the reader, e.g., for compounds 36 there is no need to report FAK IC50 as 8,57 nM and HCT IC50 as 0,004 μM, it would be more reasonable to present both in nM values.

Author Response

Answers to Reviewer #2:

  1. consistency in writing should be checked e.g. writing HeLa cells and Hela cells

Response: Thanks for your valuable comments. We feel sorry for our carelessness and have corrected it according to your valuable comments in the revised manuscript.

  1. Frequency of the use of the phrase Chen’s group .... should be reduced.

Response: Thanks for your valuable comments. We feel sorry for our carelessness and have corrected it according to your valuable comments in the revised manuscript.

  1. in several instances H, N, O, should be used in italic

Response: Thanks for your valuable comments. We feel sorry for our carelessness and have corrected it according to your valuable comments in the revised manuscript.

  1. Page 2, line 70: "Another new FAK, CEP-37440" should be corrected to "Another new FAK inhibitor, CEP-37440"

Response: Thanks for your valuable comments. We feel sorry for our carelessness and have corrected the mistake in the revised manuscript.

  1. Page 3, line 111: "targeting FAK is emerging" as there are 6 ongoing clinical trials, targeting FAK can not be emerging.

Response: Thanks for your valuable comments. We feel sorry for our carelessness and have corrected the mistake in the revised manuscript.

  1. Page 3, line 116: Table 1: size of chemical structures should be unified (e.g. all aromatic rings should be of identical size)

Response: Thanks for your valuable comments. We feel sorry for our carelessness and have corrected it according to your valuable comments in the revised manuscript.

  1. Page 9: Figure 7: correct "BTK inhibitors" to "BTK inhibitor"

Response: Thanks for your valuable comments. We feel sorry for our carelessness and have corrected the mistake in the revised manuscript.

  1. Page 10, line 263: Somebody forgot about the final revision of this paragraphs, which currently reads "Ma’s group design novel phosphamide by designed novel phosphonamide. Because these compounds are phosphonamides . in 2017 [87]"

Response: Thanks for your valuable comments. We feel sorry for our carelessness and have corrected the mistake in the revised manuscript.

  1. Page 13, line 316: listing docking energies is irrelevant here, not mentioning the accuracy (four digital places) - (Energy score: −16.5776 kJ mol−1)

Response: Thanks for your valuable comments. We feel sorry for our carelessness and have corrected the mistake in the revised manuscript.

  1. Page 16, Figure 15: Presenting numerical values should eb unified for helping the reader, e.g., for compounds 36 there is no need to report FAK IC50 as 8,57 nM and HCT IC50 as 0,004 μM, it would be more reasonable to present both in nM values.

Response: Thanks for your valuable comments. We feel sorry for our carelessness and have corrected it according to your valuable comments in the revised manuscript.

Round 2

Reviewer 1 Report

The authors have improved the manuscript based on the referee comments.

Reviewer 2 Report

Authors improved the manuscript according to the suggestions. I have notice two typing mistakes, but they will be corrected in the editing process.